# High-frequency spike inference with particle Gibbs sampling

**Giovanni Diana[1]\*, B Semihcan Sermet[1], Gerard J Broussard[2], Samuel S-H Wang[2], David A DiGregorio[1,3]\***

[1]Institut Pasteur, University of Paris, CNRS UMR 3571, Synapse and Circuit Dynamics Laboratory, Paris, France; [2]Princeton Neuroscience Institute, Princeton, United States; [3]Department of Physiology and Biophysics, University of Colorado School of Medicine, Aurora, United States

## eLife Assessment

This study presents a **valuable** contribution by introducing a model-based, Bayesian method for inferring action potentials from calcium imaging data that directly quantifies uncertainty in spike timing through posterior distributions. Using a Monte Carlo particle Gibbs sampling approach, the method achieves temporal resolution and accuracy comparable to existing techniques while offering the key added benefit of principled uncertainty estimates. The underlying methodology and characterization are **convincing**, and the work will be of particular interest to theoretically oriented neuroscientists seeking rigorous new tools for data-driven parameter inference.

**\*For correspondence:**
g.diana.mail@gmail.com (GD);
david.digregorio@cuanschutz.edu (DADiG)

**Competing interest:** The authors declare that no competing interests exist.

**Abstract** Calcium-sensitive fluorescent indicators enable monitoring of spiking activity in large neuronal populations in animal models. Despite the plethora of algorithms developed over the past decades, accurate spike-time inference methods for spike rates exceeding 20 Hz are lacking. More importantly, little attention has been devoted to the quantification of statistical uncertainties in spike time estimation, which is essential for assigning confidence levels to inferred spike patterns. To address these challenges, we introduce (1) a statistical model that accounts for bursting neuronal activity and baseline fluorescence modulation and (2) apply a Monte Carlo strategy (particle Gibbs with ancestor sampling) to estimate the joint posterior distribution of spike times and model parameters. Our method is competitive with state-of-the-art supervised and unsupervised algorithms, as evaluated on the CASCADE benchmark datasets. Analysis of fluorescence transients recorded with the ultrafast genetically encoded calcium indicator GCaMP8f demonstrates that our method can resolve interspike intervals as short as 5 ms. Overall, our study describes a Bayesian inference method for detecting neuronal spiking patterns and quantifying their uncertainty. The use of particle Gibbs samplers enables unbiased estimates of spike times and all model parameters, providing a flexible statistical framework for testing more specific models of calcium indicators.

## Introduction

Fluorescence indicators of calcium activity allow us to monitor the dynamics of neuronal populations both in vivo and in vitro. In the last decade, there has been a proliferation of new methods to identify single spikes from fluorescence time series using template matching (*Grewe et al., 2010*; *Dyer et al., 2013*; *Kerr et al., 2005*; *Quan et al., 2010*), linear deconvolution (*Holekamp et al., 2008*; *Tubiana et al., 2020*; *Wei et al., 2020*; *Yaksi and Friedrich, 2006*), finite rate of innovation (*Oñativia et al., 2013*; *Oñativia and Dragotti, 2015*), independent component analysis (*Mukamel et al., 2009*), non model-based signal processing (*Sebastian et al., 2019*), supervised learning (*Rupprecht et al., 2021*;

*Theis et al., 2016*; *Sebastian et al., 2021*; *Hoang et al., 2020*; *Sasaki et al., 2008*), constrained non-negative matrix factorization (*Zhou et al., 2018*; *Pnevmatikakis et al., 2014*; *Pnevmatikakis et al., 2016*), active set methods (*Friedrich and Paninski, 2016*; *Friedrich et al., 2017*), convex and non-convex optimization methods (*Jewell et al., 2020*; *Stern et al., 2020*; *Jewell and Witten, 2018*; *Malik et al., 2011*; *Ranganathan and Koester, 2010*), and interior point method (*Vogelstein et al., 2010*).

Model-based approaches aim to describe the mechanistic link between spikes and fluorescence. This is typically achieved by introducing a set of time-dependent 'state' variables, such as calcium level and baseline modulation, and their temporal dynamics. The temporal evolution of the state variables usually depends on additional parameters that remain constant over time and define the dynamics. Depending on the model, state variables may not be fully observable, in which case, we must infer them from the data. In statistics, this class of models is referred to as state-space models, and they are used in time series analysis to describe the probabilistic dependence between data and unobserved variables. Previous works have used tractable state-space models to derive maximum-a-posteriori estimates of spike times (*Deneux et al., 2016*; *Fletcher and Rangan, 2014*; *Kazemipour et al., 2018*; *Tsunoda et al., 2010b*).

The vast majority of spike inference methods provide single estimates of spike times by minimizing a cost function defined by the underlying model and constraints. This optimization approach does not provide information about the statistical uncertainty associated with single-point estimates. To address this issue, previous studies proposed Bayesian inference methods (*Mishchenko et al., 2011*; *Pnevmatikakis et al., 2013*; *Mishchenko and Paninski, 2011*; *Huys and Paninski, 2009*; *Vogelstein et al., 2009*; *Theis et al., 2013*; *Tsunoda et al., 2010a*; *Greenberg et al., 2018*; *Speiser et al., 2017*; *Im et al., 2019*; *Rahmati et al., 2016*; *Shibue and Komaki, 2020*), which give access to the full probability distribution of the unknowns given the data.

However, the state-space models used in these approaches do not take into account the possibility of burst firing and slow changes in the fluorescence baseline, which is known to be an important issue for the analysis of *in vivo* recordings. Moreover, current Bayesian methods do not treat time-independent model parameters (e.g. rate constants) and dynamic variables equally. Instead, they require additional optimization procedures to calibrate model parameters, typically relying on ad-hoc tuning or grid search. This separation can lead to biased inference and poorly calibrated uncertainty estimates, particularly when parameters, such as calcium decay time or spike amplitude, are inaccurately specified. In contrast, our approach jointly infers both spike times and model parameters within a unified Bayesian framework, enabling uncertainty-aware estimation and avoiding separate, error-prone calibration steps.

Inference on non-linear and non-Gaussian state-space models is analytically intractable, requiring the application of Monte Carlo methods to obtain unbiased approximations of the posterior distributions. Because the number of unknowns in these models is of the order of the number of time steps in the fluorescence time series, the analysis of longer datasets requires efficient strategies to sample from high-dimensional spaces. A major breakthrough in analyzing state-space models has been the introduction of sequential Monte Carlo methods (*Chopin and Papaspiliopoulos, 2020*). These algorithms can sample efficiently from the latent space by approximating the target posterior distribution sequentially by combining importance sampling and resampling techniques. In particular, the particle Gibbs algorithm can be used to obtain unbiased estimates of the joint distribution of time-independent model parameters and dynamical variables. Still, it has never been applied in the context of spike inference.

In this work, we employ the particle Gibbs (PG) sampler on a bursting autoregressive (BAR) model of time-series calcium-dependent fluorescence recordings to provide not only point estimates of spike times but also to quantify the statistical uncertainty of each estimate. This is important for downstream analyses, such as comparing activity across neurons or conditions. Our generative model accounts for periods of high firing rates between periods of baseline (lower) firing rates. By quantifying the performance of our method (PGBAR) on the CASCADE benchmark dataset (*Rupprecht et al., 2021*), we showed that our approach is competitive with existing techniques. Finally, we tested PGBAR on in vitro recordings of cerebellar granule cells using the ultrafast GCaMP8f calcium indicator, demonstrating that our method reliably detects spikes even at high firing rates ($\sim 200 Hz$).

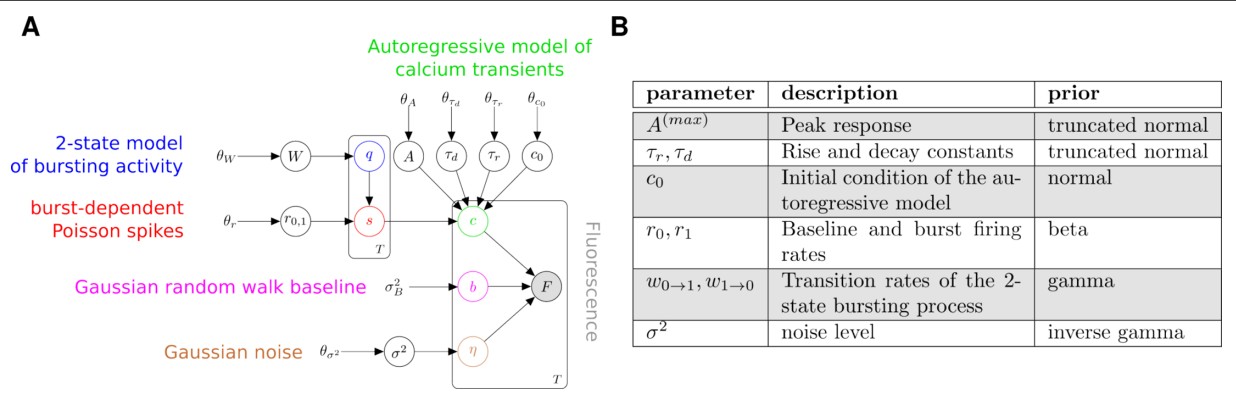

**Figure 1.** Generative model of fluorescence time-series. (**A**) Graphical representation of the generative model described in the main text. White circles denote unknown variables, gray circles denote measurements, and bare variables are fixed prior hyperparameters. Plates denote groups of variables. (**B**) List of parameters and corresponding priors.

## Results

### The model

To infer spike times and their uncertainty from noisy fluorescence traces, we first build a probabilistic generative model that captures the main dynamics underlying the fluorescence signal (**Figure 1**). We model the normalized fluorescence $F_{1:T}$ as the sum of a calcium-dependent fluorescence level $c_t$ (hereinafter referred to as calcium level for brevity), a time-varying baseline $b_t$, and Gaussian noise $\eta_t$

$$F_t = c_t + b_t + \eta_t, \qquad t = 1, \cdots, T \qquad (1)$$

where the fluorescence noise $\eta_t$ is normally distributed with zero mean and variance $\sigma^2$. During in vivo neural recordings, animal sensory stimulation, and behavior can vary. Neural activity can reflect this non-stationarity and alter firing rates over the recording period. To account for varying firing rates, our model makes the simplifying assumption that firing rates are bimodal, modeled as a hidden two-state process that separates periods of high and low firing rates. Therefore, we introduce firing states $q_t = 0, 1$, associated with low and high firing rates, $r_0$ and $r_1$, respectively. We allow for stochastic transitions between these two states with rates $w_{0\rightarrow1}$ and $w_{1\rightarrow0}$. The probability of switching from $q$ to $q'$ within a sampling period $\Delta$ is given by the transition matrix

$$W = \begin{bmatrix} 1 - w_{0\rightarrow1}\Delta & w_{0\rightarrow1}\Delta \\ w_{1\rightarrow0}\Delta & 1 - w_{1\rightarrow0}\Delta \end{bmatrix} \qquad (2)$$

The number of spikes at time $t$, $s_t$, is modeled by a Poisson distribution with rate $r_1$ when $q_t = 1$ otherwise with baseline firing rate $r_0$ when $q_t = 0$. The dynamics of the calcium level in response to a spike train is modeled as a second-order autoregressive process

$$c_t = \begin{cases} c_0 + As_1 & t = 1 \\ \gamma_1 c_1 + As_2 & t = 2 \\ \gamma_2 c_{t-2} + \gamma_1 c_{t-1} + As_t & t > 2 \end{cases} \qquad (3)$$

where $c_0$ is the initial calcium level and $A$ controls the calcium increase upon single action potential. Note that in **Equation 3** the calcium level at time $t$ depends on the previous calcium levels up to $c_{t-2}$. The dynamics of $c_t$ in response to a single spike (kernel response) is characterized by a finite rise time (time to peak response) and exponential decay (see Section Response kernel for a derivation).

We can recast our model as a first-order Markov process, where the state at time $t$ only depends on the state at time $t - 1$. This can be done by introducing a calcium vector and a spike count vector (see S2.2.3 in **Pnevmatikakis et al., 2016**)

$$C_t = \begin{bmatrix} c_t \\ c_{t-1} \end{bmatrix}, \quad S_t = \begin{bmatrix} s_t \\ 0 \end{bmatrix}, \tag{4}$$

where in particular the calcium vector at time $t$ is constructed by combining the calcium values at the current and previous time. With this definition, the calcium vector $C_t$ satisfies the first-order Markov dynamics

$$C_t = \begin{cases} [c_0 + As_1, 0] & t = 1 \\ M \cdot C_{t-1} + AS_t & t > 1 \end{cases}, \quad M = \begin{bmatrix} \gamma_1 & \gamma_2 \\ 1 & 0 \end{bmatrix}. \tag{5}$$

In this model, calcium dynamics are a deterministic process given the spike sequence $s_{1:T}$.

We reparameterize the autoregressive model in terms of phenomenological parameters: peak response ($A^{(max)}$), rise time (time to peak response $\tau_r$), and decay time ($\tau_d$) of unitary fluorescence response. Thanks to previous characterization of GCaMP probes (*Chen et al., 2013*), we can more easily design prior distributions for such phenomenological parameters rather than the autoregressive model parameters $\gamma$'s, $A$, and $\gamma_{1,2}$. $A^{(max)}$, $\tau_r$, and $\tau_d$, referred to as kernel parameters in the upcoming sections, can be derived from $\gamma_{1,2}$ and $A$ as (see Section Reparameterization for a derivation).

$$A^{(\text{max})} = A \cdot g_A, \qquad g_A \equiv \left(\frac{g+}{g-}\right)^{\frac{g+}{g-}} \left(1 - \frac{g+}{g-}\right) \left(e^{g+} - e^{g-}\right)^{-1} \tag{6}$$

$$\tau_r = \frac{\log\left(\frac{g+}{g-}\right)}{g- - g+}, \qquad g\pm = \log\left(\frac{\gamma_1 \pm \sqrt{\gamma_1 + 4\gamma_2}}{2}\right) \tag{7}$$

$$\tau_d = -\frac{1}{g+}, \tag{8}$$

Finally, the fluorescence baseline $B_t$ is described by a Gaussian random walk with a normally distributed initial condition

$$\begin{cases} B_t \sim \mathcal{N}(0, 1) & t = 1 \\ B_t \sim \mathcal{N}(B_{t-1}, \sigma_B^2 \Delta) & t > 1 \end{cases} \tag{9}$$

where $\Delta$ is the sampling period of the time series.

In the language of state-space models, the latent state of our model is the combination of the bursting state $q_t$, the spike count $s_t$, the calcium vector $C_t$ and the baseline $b_t$, whereas the fluorescence $F_t$, defined in *Equation 1* is our observation. The time-independent parameters of our model are the firing rate constants $r_{0,1}$, the transition rates of the 2-state bursting process $W_{0 \to 1}, W_{1 \to 0}$, the kernel parameters of the calcium indicators (peak amplitude, rise and decay constants), the initial calcium level $c_0$ and the fluorescence noise $\sigma$. To simplify the notation, we will denote the latent space as $X = \{q_t, s_t, C_t, b_t\}$ and the combination of time-independent parameters as $\theta$.

## State-space model formulation

The joint probability of the latent state trajectory $X_{1:T}$ and the fluorescence observations $F_{1:T}$ conditional to the time-independent parameters $\theta$ can be expressed as

$$P(X_{1:T}, F_{1:T} | \theta) = \mu^\theta(X_1) \cdot \prod_{t=2}^{T} f_t^\theta(X_t | X_{t-1}) g_t^\theta(F_t | X_t) \tag{10}$$

where $f_t^\theta(X_t | X_{t-1})$ is the transition probability of the latent state, $g_t^\theta(F_t | X_t)$ is the probability of the observed fluorescence conditional on the latent state at time $t$ and $\mu^\theta(X_1)$ is the probability distribution of the initial latent state. The latent state transition probability can be expressed in terms of the calcium level, firing state, and baseline transitions and the Poisson probability of spike counts, namely

$$f_t^\theta \left( X_t \mid X_{t-1} \right) \quad = \overbrace{\delta^{(2)} \left( C_t - M \cdot C_{t-1} - AS_t \right)}^{\text{deterministic calcium}} \cdot \overbrace{W_{q_{t-1} q_t}}^{\text{firing state}} \cdot \overbrace{\frac{\left( r_{q_t} \Delta \right)^{s_t}}{s_t!} e^{-r_{q_t} \Delta}}^{\text{Poisson spikes}} \cdot$$

$$\overbrace{\left( 2\pi \Delta \sigma_b^2 \right)^{-1/2} \exp \left( -\frac{1}{2\Delta \sigma_b^2} \left( b_t - b_{t-1} \right)^2 \right)}^{\text{baseline}} \cdot \tag{11}$$

By assuming the fluorescence noise to be normally distributed, we have

$$g_t^\theta (F_t | X_t) = (2\pi \sigma^2)^{-1/2} \cdot \exp \left[ -\frac{1}{2\sigma^2} (F_t - c_t - b_t)^2 \right]. \tag{12}$$

To infer latent states and time-independent parameters from fluorescence observations, we need to compute the posterior probability

$$P(X_{1:T}, \theta | F_t) = \frac{P(\theta) \cdot P(X_{1:T}, F_{1:T} | \theta)}{P(F_{1:T})}, \tag{13}$$

where $P(\theta)$ denotes the prior probability on model parameters and $P(F_{1:T})$ is the normalization factor of the posterior distribution, also known as marginal likelihood. This distribution encodes all the information about the statistics of the latent state trajectory and the model parameters. We can use it to compute point estimates but also to quantify uncertainties. The posterior distribution for general state-space models is not analytically tractable. However, we can use Monte Carlo methods to generate random samples from the posterior distribution and use them to obtain unbiased approximations of any posterior average.

## Sequential Monte Carlo

The model described in the previous section is analytically intractable; therefore, we employed Monte Carlo methods to sample from the posterior distribution in *Equation 13* of spike times and model parameters, allowing us to make probabilistic inferences rather than relying on point estimates alone. There are two critical issues when applying these methods to state-space models. First, the high-dimensionality of the posterior distribution and, second, the joint inference of state variables and constant parameters. In the general setting of a state-space model, we need to estimate state variables at each time point. Consequently, the support of the target posterior distribution is a high-dimensional space. The high dimensionality of state-space models poses a challenge when applying standard sampling methods, such as Markov Chain Monte Carlo, because their performance degrades rapidly with increasing dimensionality. Sequential Monte Carlo (SMC) methods address this issue by providing efficient sampling strategies from the latent space. The typical approach is to construct a sequential approximation to the posterior distribution, in which observations are accounted for iteratively. SMCs solve the problem of estimating dynamical variables at fixed model parameters (filtering).

The second issue, related to the joint estimation of parameters and dynamic variables, was addressed by *Andrieu et al., 2010* with the introduction of the particle Gibbs algorithm. This algorithm alternates the sampling of model parameters and state variables as in the Gibbs sampler, with the difference that state variables are sampled from an SMC-based transition kernel that leaves the filtering distribution $P(X_{1:T} | F_{1:T}, \theta)$ invariant. In this work, we employ a version of the particle Gibbs algorithm developed by *Lindsten et al., 2014*, named particle Gibbs with ancestor sampling (PGAS), with better mixing properties (see Algorithm 2 and the Methods section).

To carry out inference of time-independent and dynamical variables for our model, we employed Algorithm 1, which alternates the two steps mentioned above: (1) run the PGAS transition kernel to sample a new trajectory of the state variables, and (2) sample model parameters from their full conditional distributions when analytically available, otherwise from a Metropolis-Hastings kernel.

Algorithm 1. **Gibbs sampler.**

1: Set $\theta^{(1)}$ and $X_{1:T}^{(1)}$
2: **for** $n > 1$ **do**
3:  draw $X_{1:T}^{(n)} \sim \mathcal{K}_{\theta^{(n-1)}}^{N}(X^{(n-1)}, \cdot)$ (PGAS kernel)
4:  draw $\theta^{(n)} \sim P(\theta|X_{1:T}^{(n)}, F_{1:T})$

## Validation and performance of PGBAR on simulated time-series fluorescence data

To test the performance of our inference method, we generated latent state variables and fluorescence time series from our model, and compared the spikes inferred using our sampling algorithm against the ground-truth simulations. In *Figure 2A*, we show a fluorescence time series simulated from our model. The firing pattern exhibits periods of increased firing rate interspersed with quiet time windows. By using this trace as input to Algorithm 1, we can generate a latent state trajectory $X_{1:T} = \{q_t, s_t, C_t, b_t\}_{t=1}^{T}$ and a set of model parameters at each iteration. In *Figure 2B*, we show 1000 samples of spike counts obtained by fitting the normalized fluorescence in *Figure 2A*. The average spike counts over the random samples at each time frame (*Figure 2C*) can be interpreted as the instantaneous firing rate multiplied by the sampling period. To illustrate the accuracy of our method, we calculated the spike counts within 1 s time intervals for each random sample, yielding the posterior distribution of the number of spikes in each time bin. As shown in *Figure 2D*, the ground-truth spike counts are well within the range of the posterior. Our method allows us to infer not only spike times but also the time windows of high and low firing states ($q_t = 0, 1$ in the model) of the neuron. In particular, the probability of a burst-firing state ($q = 1$) can be obtained by averaging the firing state across the Monte Carlo samples. As shown in *Figure 2E*, this probability is close to one during the ground truth bursting periods and zero otherwise, with some degree of uncertainty at the onset and offset of the bursting period. *Figure 2F* compares the ground-truth baseline and the sample average.

One of the key advantages of our sampling algorithm is the joint estimation of latent states and time-independent model parameters, such as spike amplitude, decay time, noise level, and baseline variance. This contrasts with most existing spike inference algorithms, which rely on fixed or externally calibrated parameters. Such fixed-parameter methods are vulnerable to systematic errors when parameter values are uncertain or poorly estimated. By jointly sampling from the posterior over all variables and parameters, our method propagates uncertainty correctly and mitigates biases arising from manual tuning or poor initialization.

*Figure 2G* illustrates the posterior distributions associated with the fluorescence probe's peak amplitude, noise level, decay, and rise time. The ground-truth parameters used to simulate the time series traces are always close to the peak of the corresponding posterior distributions, showing that our model is identifiable.

To quantify the importance of having two firing states on the accuracy of the inference, we compared the performance of our method against a variant with only one global Poisson firing rate. We simulated fluorescence traces at different signal-to-noise (SNR) levels (1.1, 2, and 10), defined as the ratio between peak response $A^{(max)}$ and the fluorescence noise parameter $\sigma$, and the burst firing frequency parameter, $r_1$ (5 Hz, 10 Hz, 20 Hz, 50 Hz). Then, we used our algorithm and its non-bursting variant to infer spikes from the fluorescence trace. To quantify the inference performance, we calculated the correlation between the ground truth and the estimated spikes (downsampled at 7.5 Hz for consistency with other analyses in this work), the average absolute error and the bias (average error) per time point (see Methods). The top panel of *Figure 3A* shows two example traces with stimulation times generated by a 5 and 50 Hz Poisson distribution of spike times. For the 5 Hz firing trace, the bursting model did not improve the inference accuracy compared to the variant with a single firing rate (*Figure 3A*, middle and bottom). The two analyses produced comparable correlations, errors, and biases (*Figure 3C–E*). In contrast, the single-firing-rate model produced a systematic underestimation of the number of spikes for the 50 Hz trace, whereas the original model reliably captured the ground-truth spikes.

The lower performance of the non-bursting version of the model is due to the bias induced by forcing a single firing rate across the time series. *Figure 3E*While for all noise and frequency

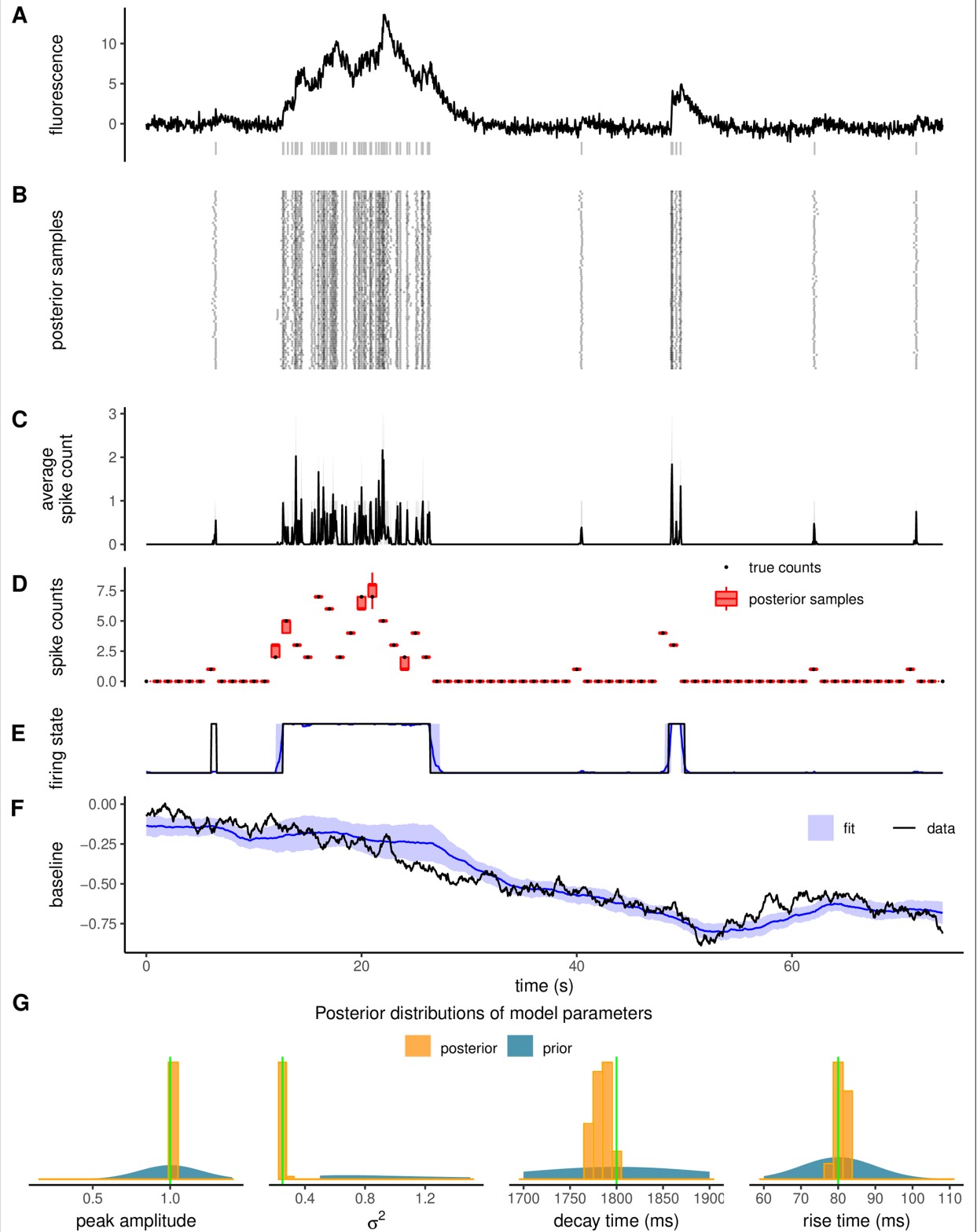

**Figure 2.** Validation of the spike inference approach with simulated data. (**A**) Example trajectory simulated from the model (solid, black) with ground truth spike times shown underneath (gray vertical lines). (**B**) Raster plot representing spike times for a thousand Monte Carlo samples. (**C**) Average spike counts over the Monte Carlo samples at each time frame. (**D**) Comparison between ground-truth counts over 1 s bins (black dots, from the example trace in A) and the corresponding posterior distributions (red boxes). (**E–F**) Comparison of ground truth firing state and baseline (solid, black) to estimated ones (blue). Shading indicates one standard deviation from posterior averages. (**G**) Posterior distributions of peak response upon a single spike, decay time, rise time, and noise level compared with the true value (vertical lines in green).

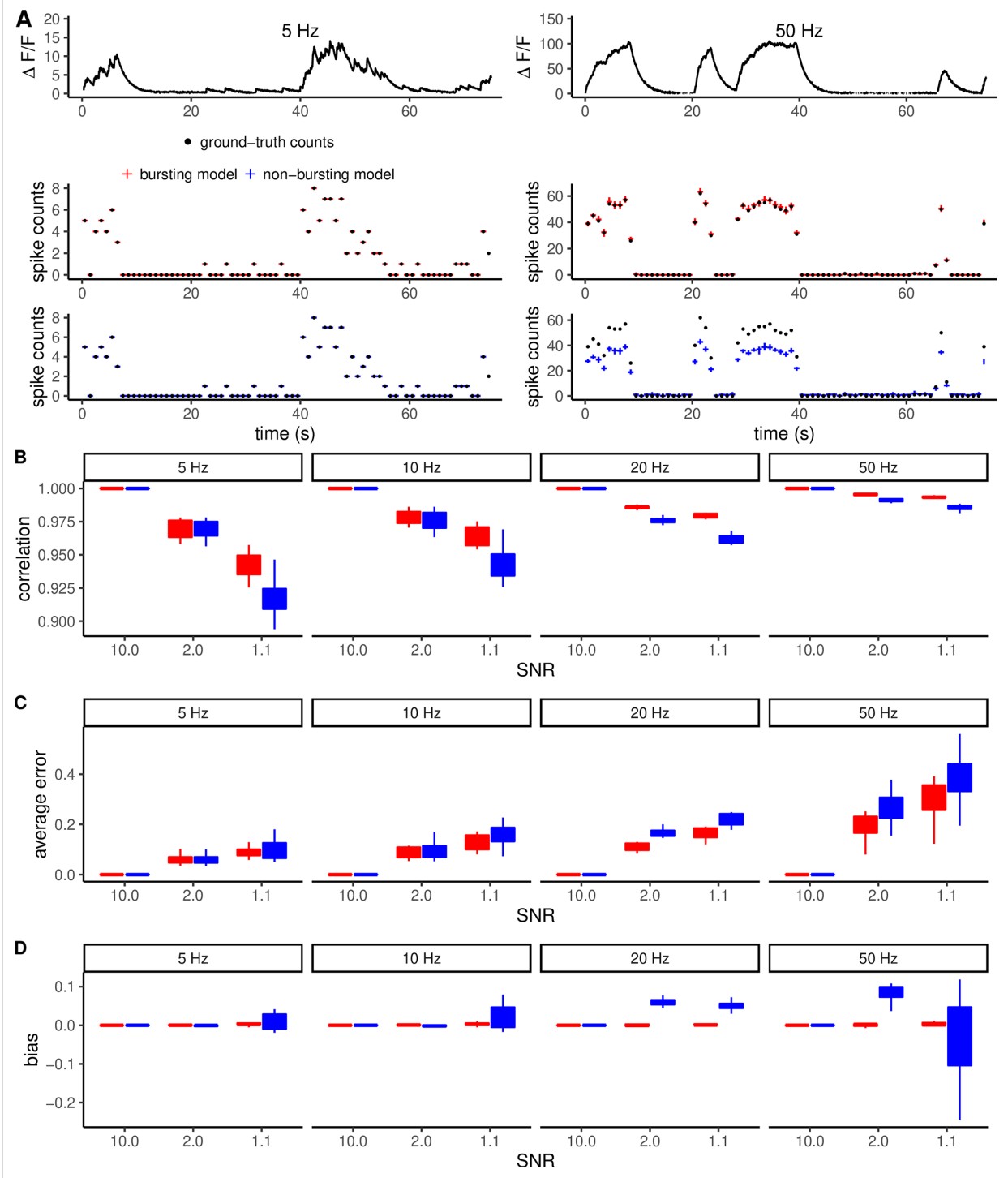

**Figure 3.** Dependency of inference performance on noise and firing frequency and the bias of non-bursting models. (**A**) Example fluorescence time series simulated at 5 Hz and 50 Hz bursting frequencies (top). The analysis of these traces using the model's bursting and non-bursting variants highlights the substantial bias introduced by the non-bursting variant at high frequency (bottom). (**B–D**) Quantification of correlation with true spikes, average error, and bias for different signal-to-noise ratios (SNR) and frequency. As firing frequency increases, the correlation with ground-truth spikes generally increases. This is an effect of calculating correlations at fixed temporal resolution. The average error was computed as the sum of absolute deviations from the true spike counts divided by the number of time steps.

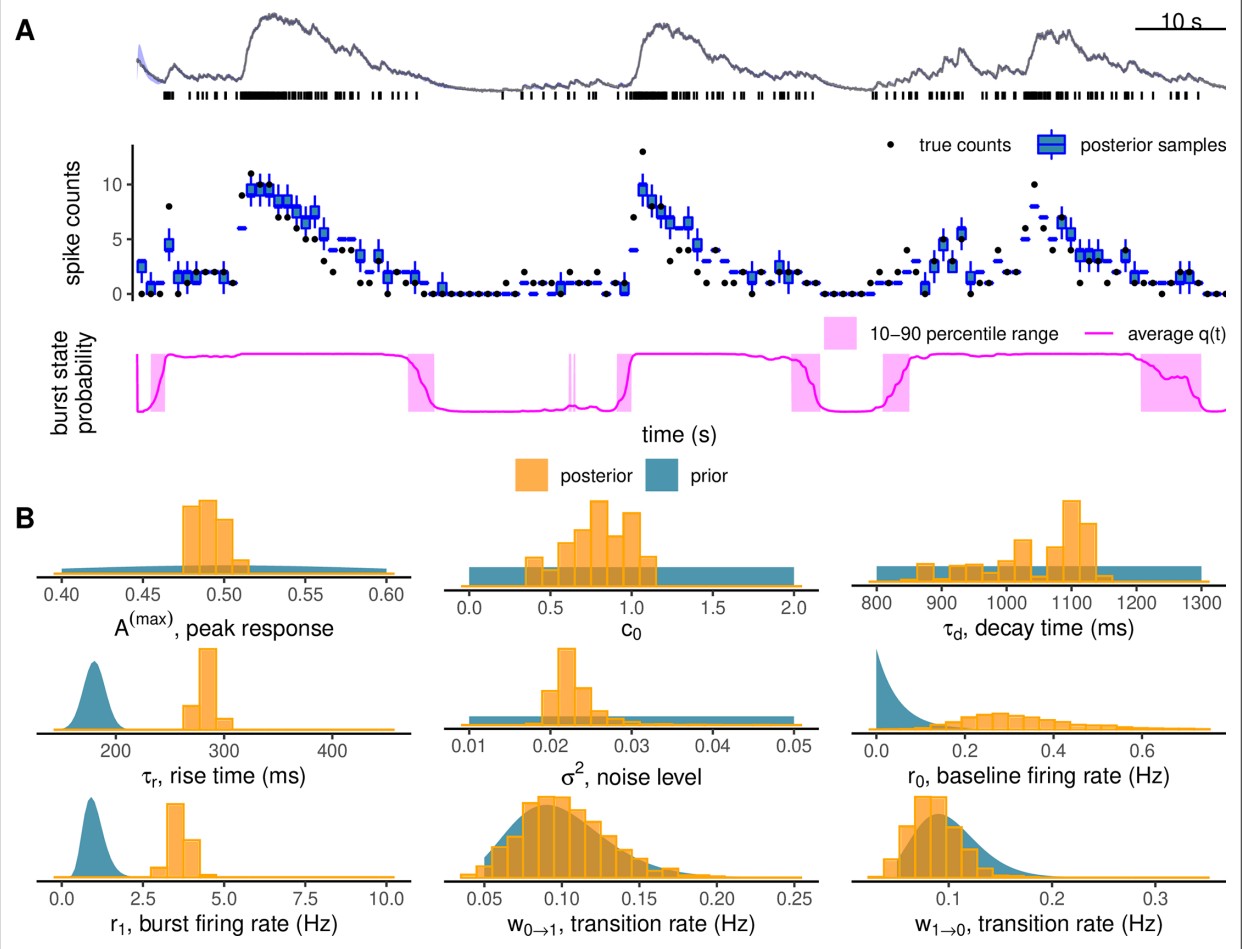

**Figure 4.** Comparison of PGBAR with existing methods using analysis of CASCADE benchmark data. (**A**) Correlation between estimated and ground-truth firing rates filtered with a 200 ms bandwidth Gaussian kernel (CASCADE dataset). The color code represents the different calcium indicators used in each dataset.(**B**) Correlation with ground-truth spikes as a function of the standardized noise level (*Rupprecht et al., 2021*). (**C**) Comparison with existing methods. Correlation averaged across datasets and neurons.

conditions, inference using the bursting model yields unbiased spike counts (Fig. 3E), in the case of the non-bursting model, the single-Poisson firing rate necessarily leads to an underestimation of the spike count during bursting time windows and an overestimation during low-activity windows. At increasing noise levels and firing frequency, the performance difference between bursting and non-bursting versions of our algorithm becomes more pronounced (*Figure 3D*), with a clear advantage of our original bursting model in increasing correlation with ground truth and reducing error.

## Validation of PGBAR on the CASCADE benchmark data and comparison to previous methods

To test our method on experimental data, we analyzed neuronal recordings from the CASCADE benchmark dataset (*Rupprecht et al., 2021*), thereby quantifying our algorithm's performance across different calcium indicators. Bayesian priors for all PGBAR parameters were adapted to each experiment based on the published characterization of the different calcium indicators used (*Chen et al., 2013*). We compared our method to CASCADE (*Rupprecht et al., 2021*), MLSpike (*Deneux et al., 2016*), Peeling (*Lütcke et al., 2013*), CaImAn (*Giovannucci et al., 2019*), Suite2p (*Pachitariu et al., 2017*), and JewellWitten (*Jewell et al., 2020*) by using their previously benchmarked performance on the same datasets obtained from extensive parameter optimization (*Rupprecht et al., 2021*; *Figure 4C*). As a metric for inference performance, we used Pearson's correlation coefficient between ground truth and predicted spikes after filtering with a Gaussian kernel with a 200 ms bandwidth. This

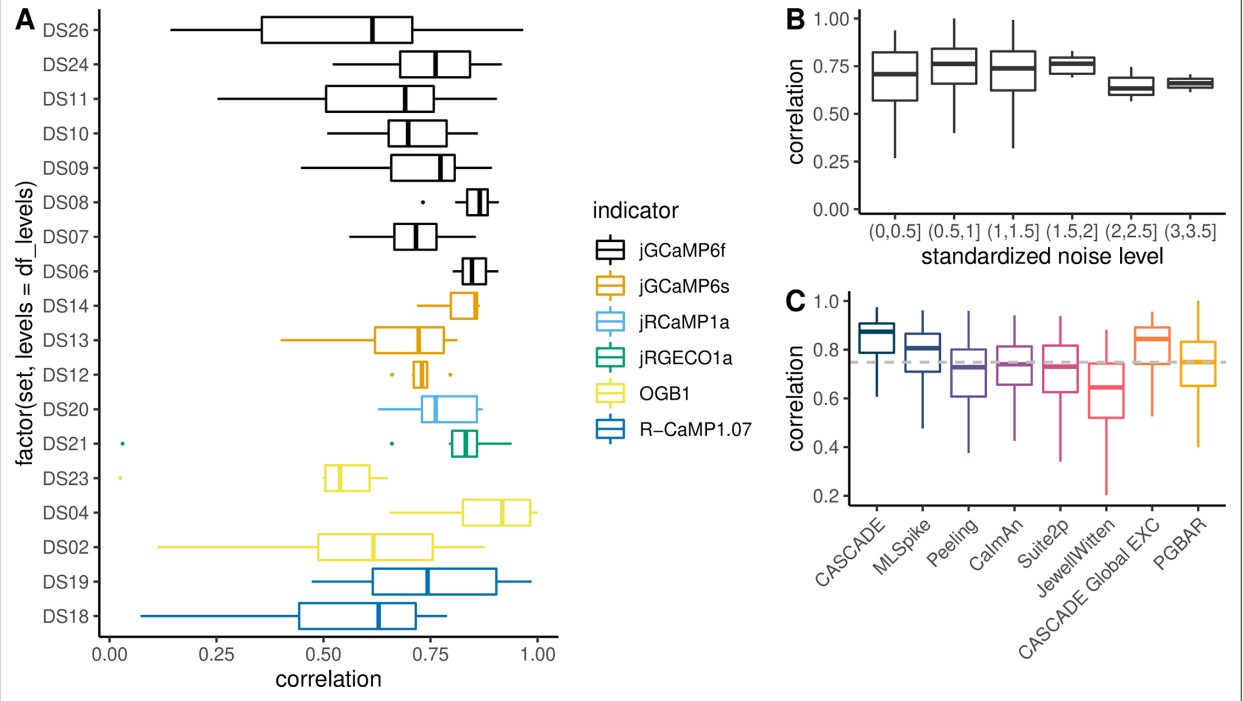

**Figure 5.** Analysis of GCaMP6f recordings from the CASCADE dataset. (**A**) Example $\Delta F/F$ from the CASCADE dataset (#DS09, GCaMP6f, mouse visual cortex) with ground-truth spikes shown underneath fluorescence (top), comparison of spike counts within 1 s time intervals (middle), and burst probability (bottom). Shading denotes uncertainty within one standard deviation. (**B**) Comparison between posterior distributions of the model parameters (histograms) and priors (continuous densities): maximal calcium response to single spikes ($A_{max}$), initial calcium level ($c_0$), decay and rise time, noise level ($\sigma^2$), bursting ($r_1$) and baseline ($r_0$) firing rates, transition rates between firing states ($w_{0\to1}, w_{1\to0}$).

metric allowed us to directly compare PGBAR with previous analyses *Rupprecht et al., 2021*. The correlation obtained with PGBAR averaged across cells and datasets is 0.75. As for the other methods, we observed a large variability of performance across recordings, however, we did not find a particular condition where our method performed systematically better or worse across indicators (*Figure 4A*) and standardized noise level (*Figure 4B*), which is a noise index introduced by *Rupprecht et al., 2021* to account for different sampling frequencies (ratio between the standard deviation of the normalized fluorescence and the square root of the sampling frequency). In particular, we did not observe a statistically significant difference in the mean correlation coefficient across standardized noise levels from 0.5 to 3.5 (*Figure 4B*). However, lower noise levels seem to be associated with a larger range of correlation.

The performance of PGBAR across the CASCADE database was comparable to that of previous methodologies. Among model-based approaches, our method slightly underperformed MLSpike (PGBAR median within 6% of MLSpike; Mann-Whitney test, *p*-value = 0.0043), likely due to its description of the nonlinear properties of the calcium indicator. The best performance is achieved by the supervised CASCADE method; however, it is a supervised method (i.e. requires training on ground-truth data) and does not provide posterior distributions for spike times and model parameters.

The previous analysis provides an overview of PGBAR's overall performance compared with prior methods. To illustrate the advantages of PGBAR in estimating statistical uncertainties, we now take a closer look at a representative recording in the CASCADE dataset (*Figure 5A*, CASCADE dataset 9 *Chen et al., 2013*), a GCaMP6f fluorescence time series from a pyramidal neuron in the mouse visual cortex. The comparison between ground-truth spikes and those inferred using PGBAR in *Figure 5A* shows differences outside the first-third interquartile range in 30% of the 1 s time intervals. This discrepancy between the posterior distributions and the ground truth can be attributed to the autoregressive model's limited capacity to capture the full biophysical properties of GCaMP6. Despite such a discrepancy, the estimated bursting pattern shown in *Figure 5A* (lower panel) faithfully captures the overall periods of increased neuronal activity . The posterior distributions of model parameters are

shown in *Figure 5B*. Some distributions (burst firing rate and the rise time) shift significantly from their corresponding priors. The mismatch between the prior and the posterior also highlights the model's shortcomings. In this case, the posterior distribution of the burst firing rate appears to be shifted to larger values. This result could be attributed to the use of Gaussian noise in the model. Indeed, non-Gaussian fluorescence fluctuations might be misinterpreted by the sampler as true spikes, thereby increasing the burst firing rate. This example illustrates that prior distributions can partially compensate for this bias by penalizing unrealistic parameter configurations.

## Validation and performance of PGBAR on simulations of short-interval stimuli

We tested the accuracy of PGBAR in resolving pairs of stimuli with inter-spike intervals (ISIs) below 10 ms by simulating the model at different signal-to-noise ratios. *Figure 6A* shows two simulated fluorescence traces with two spikes 10 ms apart at low (1.4) and high (3.4) SNR levels. When applied to these simulated data, PGBAR accurately detected two spikes with over 95% confidence across all SNR values. Next, we examine the temporal accuracy of ISI estimation. Increasing SNR or sampling frequency has the effect of narrowing the ISI posterior distribution around the ground-truth value (*Figure 6B*). When examining multiple stimulus intervals, SNRs, and sampling frequencies, we show that high sampling frequencies enable more reliable extraction of spike intervals. The posterior probability of the ISI falls within 3 ms of the ground-truth ISI at a sampling frequency of 1 kHz and within 6 ms for 3 kHz (averaging over 30 trials, see *Figure 6C*). When this probability is above 0.5, the ground-truth value is equal to the mode of the posterior ISI distribution (using 3 ms bin size), therefore, we can use this probability as a metric to quantify the temporal accuracy of our inference. For ground-truth values above 3 ms, the contours of constant probability (iso-probability) show that increasing the temporal separation between stimuli does not affect the inference accuracy. On the contrary, increasing the sampling frequency shifts the iso-probability contours toward lower SNR, indicating that higher sampling rates require lower SNR to achieve the same accuracy. *Figure 6D* shows the posterior ISI distributions obtained from ten independent simulations with ground-truth ISI of 5 ms at different SNR levels. At low SNR, the posterior distributions have larger variance, whereas at higher SNR, they shrink around the ground-truth value. This analysis illustrates the expected variability in spike inference across multiple trials of the same neuron. Based on this analysis, we expect PGBAR to provide accurate estimates of interspike intervals down to 5 ms.

## PGBAR spike inference from fluorescence transients recorded using the fast calcium indicator GCaMP8f

We tested our approach on the fast calcium indicator GCaMP8f by performing high-speed ($\approx 2.8$ kHz) two-photon linescan calcium imaging of cerebellar granule cells in vitro. GCaMP8f was expressed in the Crus I region of the cerebellum using adeno-associated virus (AAV) injection (*Figure 7A*). Compared with GCaMP6f, GCaMP8f exhibits a rise time nearly an order of magnitude faster, which we expected to translate into substantially improved temporal accuracy for spike-time detection. Fluorescence signals were recorded from both granule cell somata and synaptic boutons along the parallel fibers, while ground-truth spikes were evoked via extracellular stimulation of granule cell axons in the molecular layer (*Figure 7B*).

For each recorded soma and bouton, we applied two types of stimuli. Single time point stimulation and a fixed stimulation pattern generated from a 20 Hz Poisson process with 29 stimulation time points. First, we used the single-stimulation trials to design prior distributions of amplitudes, rise, and decay constants (*Figure 7C*). Next, we used PGBAR to analyze each Poisson stimulation trial in *Figure 7E*. By generating thousands of posterior samples of spike time patterns, we obtained the spike probability for all time frames and trials (*Figure 7F*). The Pearson's pairwise correlation between spike probabilities (Gaussian filtered with 20 ms bandwidth) across trials is always larger than 0.95, which demonstrates that PGBAR provides robust predictions across trials and can reliably detect single-trial action potential-evoked GCaMP8f fluorescence transients.

To illustrate the temporal accuracy of PGBAR, we focused on a part of the stimulus train with a short 5 ms interval between two spikes (*Figure 7G*). Despite the relatively low SNR ($A^{(max)}/\sigma \approx 2.4$), 100% of the posterior samples contained two spikes in the considered time interval. The ground-truth ISI is well within the posterior distribution of each trial, with posterior modes symmetrically distributed

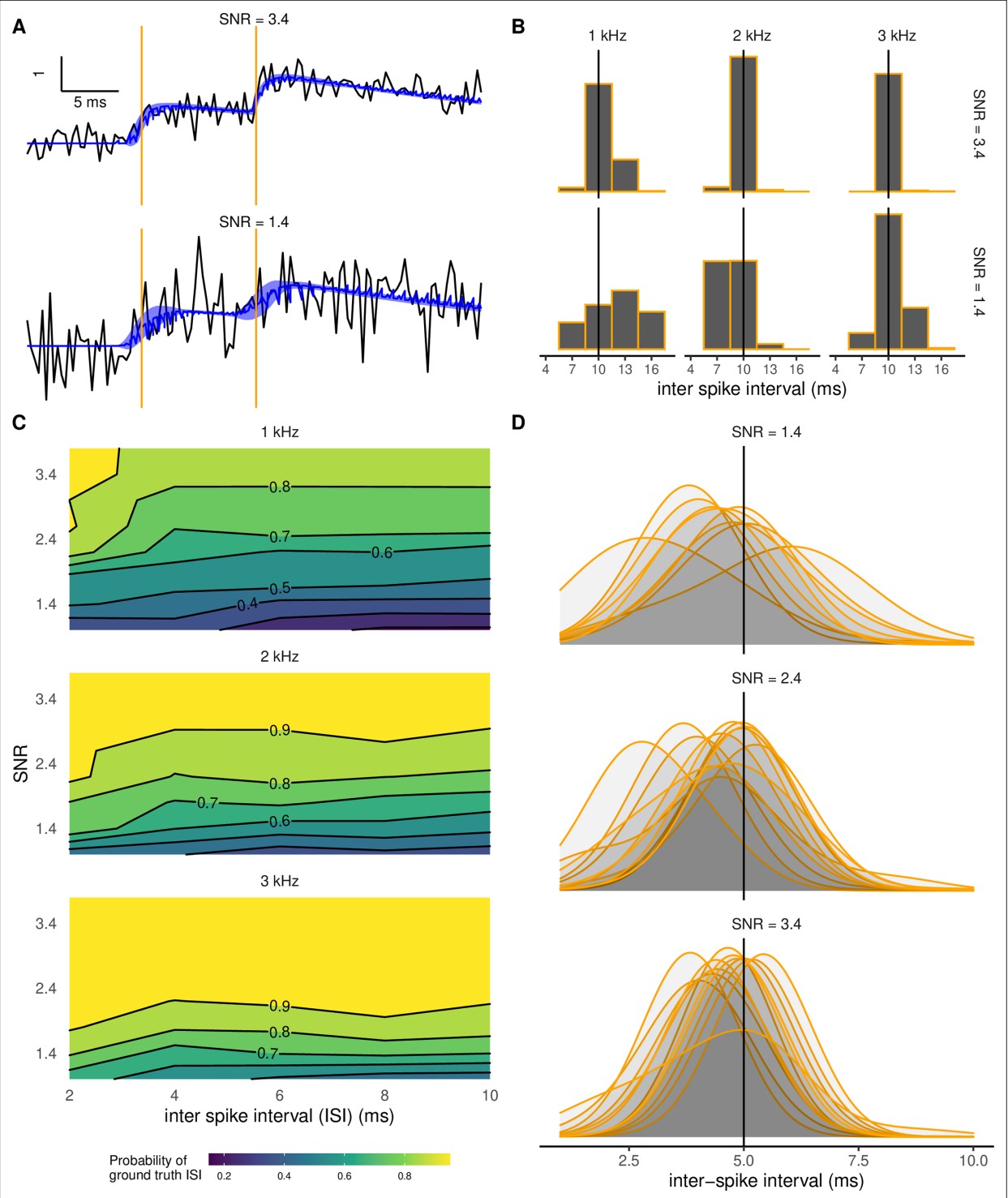

**Figure 6.** Sensitivity of spike detection to sampling frequency and signal-to-noise (SNR) level. (**A**) Examples of simulated fluorescence traces with two spikes separated by 10 ms (vertical lines) at low SNR (1.4, bottom) and high SNR (3.4, top). Shaded bands display denoised fits (calcium fluorescence plus baseline) within one standard deviation. (**B**) Posterior distributions of the inter-spike interval (ISI). Increasing SNR (from bottom to top) and sampling frequency (left to right) has the effect of reducing the inter-spike interval (ISI) posterior variance, bringing the maximum a posteriori estimate (MAP) closer to the ground truth. (**C**) Posterior probability of the ISI to be within an interval of 3 ms centered around the ground-truth ISI as a function of SNR and ground-truth ISI with sampling frequency of 1, 2, and 3 kHz. (**D**) Trial-to-trial variability of ISI posterior distributions. We analyzed 12 simulated fluorescence traces with a sampling frequency of 3 kHz and two spikes separated by 5 ms. Density plots have been smoothed with a 1 ms bandwidth. For all simulations, we used $\tau_r = 3.7$ ms and $\tau_d = 40$ ms.

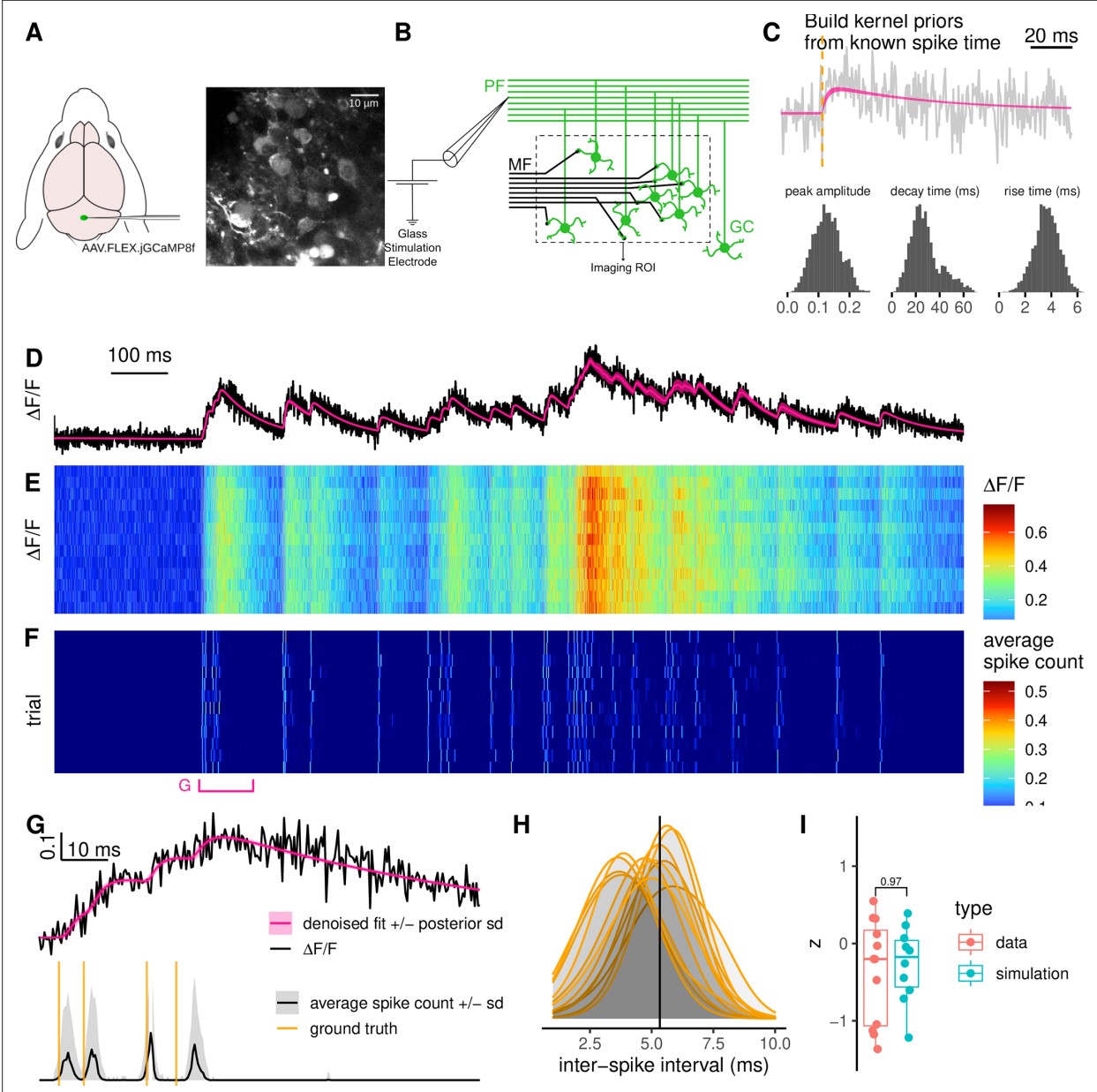

**Figure 7.** High-speed 2-photon linescan calcium imaging. (**A**) Schematic showing GCaMP8f virus injection in the cerebellar vermis. (**B**) Schematic showing the configuration of antidromic extracellular stimulation of the parallel fibers to evoke responses in GC somata or axonal boutons. (**C**) Example soma fluorescence response to a single AP stimulation and the estimation of the model parameters for the GCaMP8f indicator from all twelve trials. (**D–E**) High-speed (3 kHz) 2-photon linescan calcium imaging of granule cell somata. (**D**) Representative fluorescence time series from a single trial and (**E**) heatmap showing fluorescence transients evoked using the same Poisson train across twelve trials. Single-trial fluorescence (**D**) and denoised fit (calcium level plus baseline). (**F**) Spike detection for each trial. (**G**) 100 ms time window highlighting the first four stimulation-induced action potentials. Normalized fluorescence and denoised fit (top), average spike count (bottom). Orange vertical lines denote stimulation time points. (**H**) Comparison of the posterior distributions of the interval between the first two detected spikes across experimental trials. The solid vertical line at 5.3 ms denotes the time interval between the first two stimulations. (**I**) Comparison of the posterior distribution modes of the inter-spike interval across trials between real data and simulations.

around it (*Figure 7H*). To better understand the variability of the posterior modes, we simulated 10 fluorescence traces with two spikes separated by 5 ms and using a sampling frequency matching our recordings. By analyzing these simulations with PGBAR, we obtained a distribution of ISI posterior modes that is statistically compatible with the real data (*Figure 7I*, Mann-Whitney test *p*-value = 0.97),

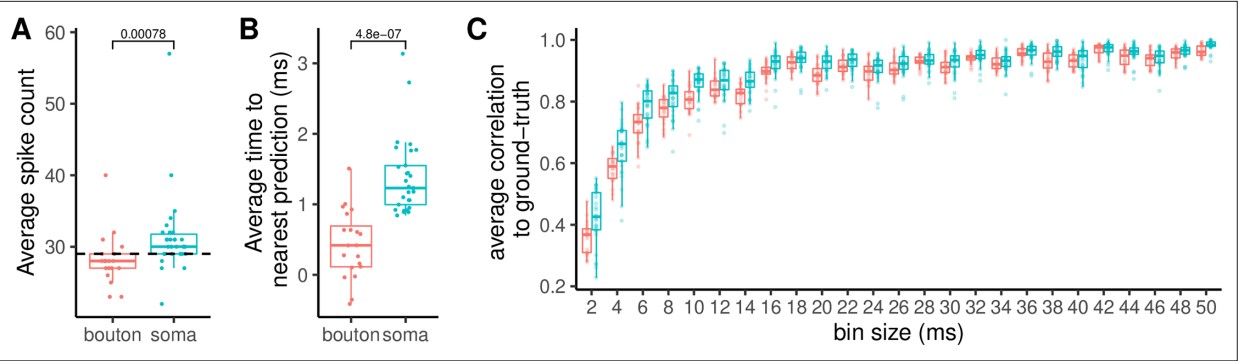

**Figure 8.** Temporal resolution analysis across soma and synaptic boutons along the parallel fiber. (**A**) Average spike count estimated from six somata and four boutons (five trials each). The dashed line denotes the number of stimulation time points (29). (**B**) Time from stimulus time predicted spike averaged across all 29 AP stimuli for each trial. The comparison between somata and boutons shows that somatic transients are delayed by 1 ms. (**C**) Correlation between predicted spikes and stimulation events across time scales.

which provides further evidence that our results are statistically unbiased and that PGBAR can be used to infer spike times for intervals down to 5 ms.

The distribution of the posterior modes of the total number of spikes across experiments and trials is centered around the ground-truth value for both somas and axonal boutons with a relative error of 15% (*Figure 8A*). To quantify the precision of spike-time detection, we used a single-trial definition of temporal accuracy as the offset between ground-truth spikes and their nearest detections averaged across stimulation time points. This temporal accuracy is 0.43 ms (±0.10 ms, SEM) and 1.39 ms (±0.11 ms, SEM) in boutons and somas, respectively, highlighting a significant delay of 0.95 ms (±0.15 ms SEM, Mann-Whitney test p-value = $4.8 \times 10^{-7}$) from the average time of detection in boutons (*Figure 8B*). This result is compatible with the proximity of synaptic boutons to the electrical stimulation along the parallel fiber. We analyzed signals from both somata and axonal boutons because in vivo two-photon imaging can be performed from both parts of the cell. Here, we showed that our method performs reliably across both recording sites, demonstrating its robustness.

As an alternative estimate of temporal accuracy, we have calculated Pearson's correlation between detected spike times and stimulation times over different time scales. We binned detected spike times and the Poisson stimulation train using bin sizes ranging from 2 ms to 50 ms and calculated their correlation. If we use the CASCADE dataset average correlation (0.75) as the reference, this analysis shows that the correlation exceeds 0.75 for bin sizes up to 10 ms, confirming that our method accurately infers short intervals.

Next, we compared the temporal accuracy of PGBAR, CASCADE, and MLspike predictions (*Figure 9A*). The temporal shifts in the estimated spike time from ground truth across MLspike, Cascade, and PGBAR have different distributions, with 10th-to-90th percentile ranges of 14 ms, 8 ms, and 3 ms, respectively. In addition, the false detection rate, defined as the difference between detected and ground-truth number of spikes divided by the ground truth, is significantly lower in PGBAR, compared to MLspike and CASCADE (Mann-Whitney test p-value $< 10^{-3}$) which show a much broader distribution across trials (*Figure 9B*). These results demonstrate that PGBAR estimates are more reliable than MLspike and CASCADE across trials, with a narrower, unbiased distribution of the average time to the nearest spike and the false-positive rate. Next, we compared the correlation with ground truth across different time scales for the three methods (*Figure 9C*). In our cerebellar dataset, PGBAR outperforms the other methods across the full range of bin sizes, particularly at short time scales, further highlighting its advantage for high firing rates.

To quantify how the hyperparameters of the priors affect PGBAR predictions, we conducted a biparametric sensitivity analysis and compared it with MLspike. For PGBAR, we considered the hyperparameters of the Bayesian priors governing the peak response to a single spike and the baseline variance, which determines the extent to which fluorescence is attributable to baseline modulation. For MLspike, we considered the transient amplitude and the decay time constant. For both methods, we varied the parameters between –50% and +50% of their optimal values. We estimated the correlation

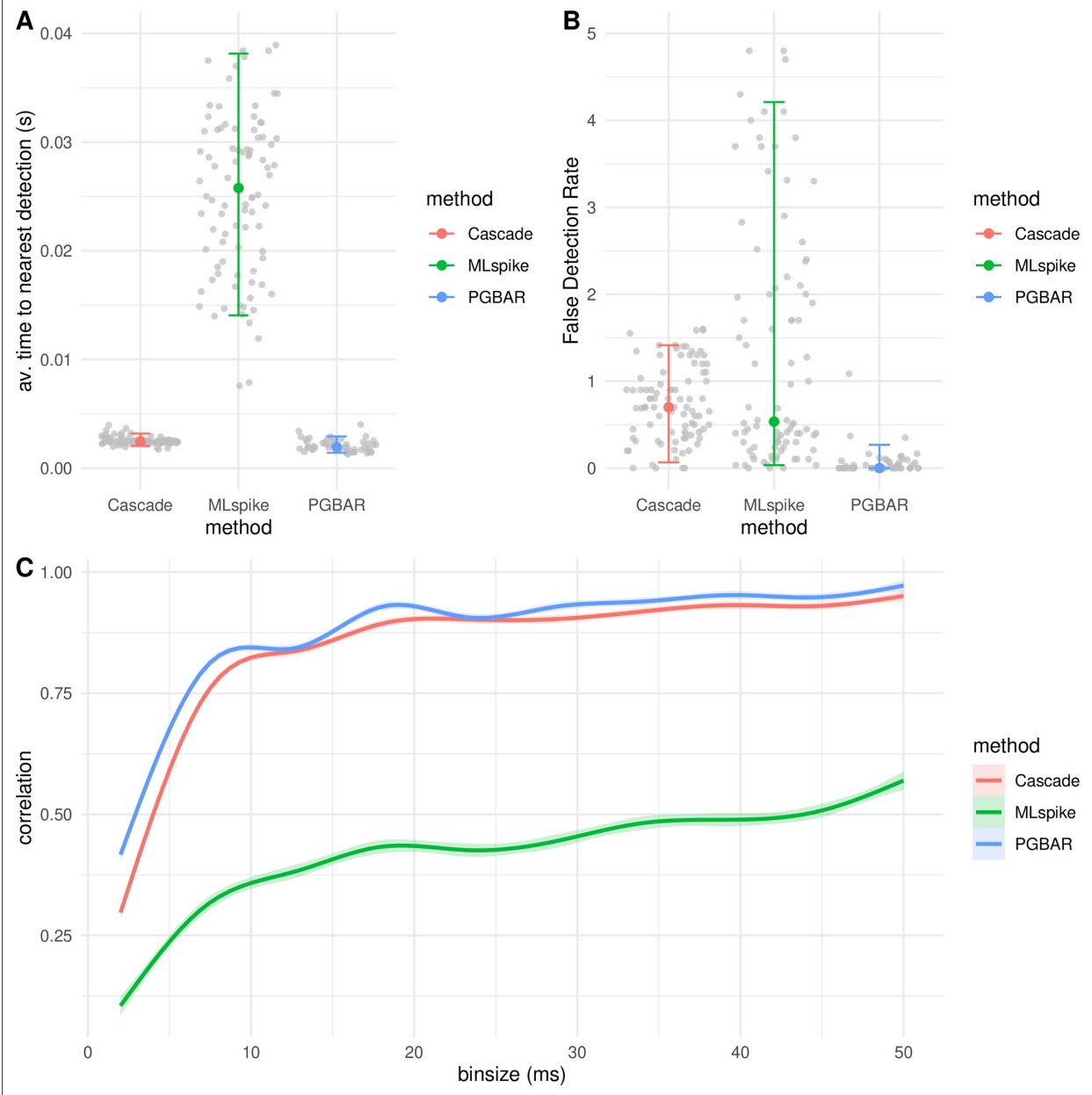

**Figure 9.** Comparison of temporal accuracy across methods. (**A**) Estimate of temporal accuracy (absolute time difference between ground truth spikes and nearest detections averaged across detections) for all granule cell somata (six somata, five trials per somata, 29 AP per trial) recordings and across inference methods (CASCADE, MLSpike, and PGBAR). (**B**) False detection rate defined as the number of excess spikes divided by the number of ground-truth spikes. (**C**) Pearson's correlation across temporal scales by method. Shading is the standard error of the mean. (**A, B**) Error bars denote the 10th-to-90th percentile range.

between predictions and ground truth as well as the number of spikes (*Figure 10A*). Both distributions of Pearson's correlation and spike number arising from different choices of parameters (model parameters for MLspike and prior hyperparameters for PGBAR) are much narrower in the case of PGBAR. To quantify this difference, we calculated the coefficient of variation across all parameter configurations for each trial (*Figure 10B*), which was significantly lower in PGBAR than in MLspike. Our analysis shows that PGBAR predictions are much less sensitive to inaccurate prior hyperparameter choices than MLspike is to its model parameters.

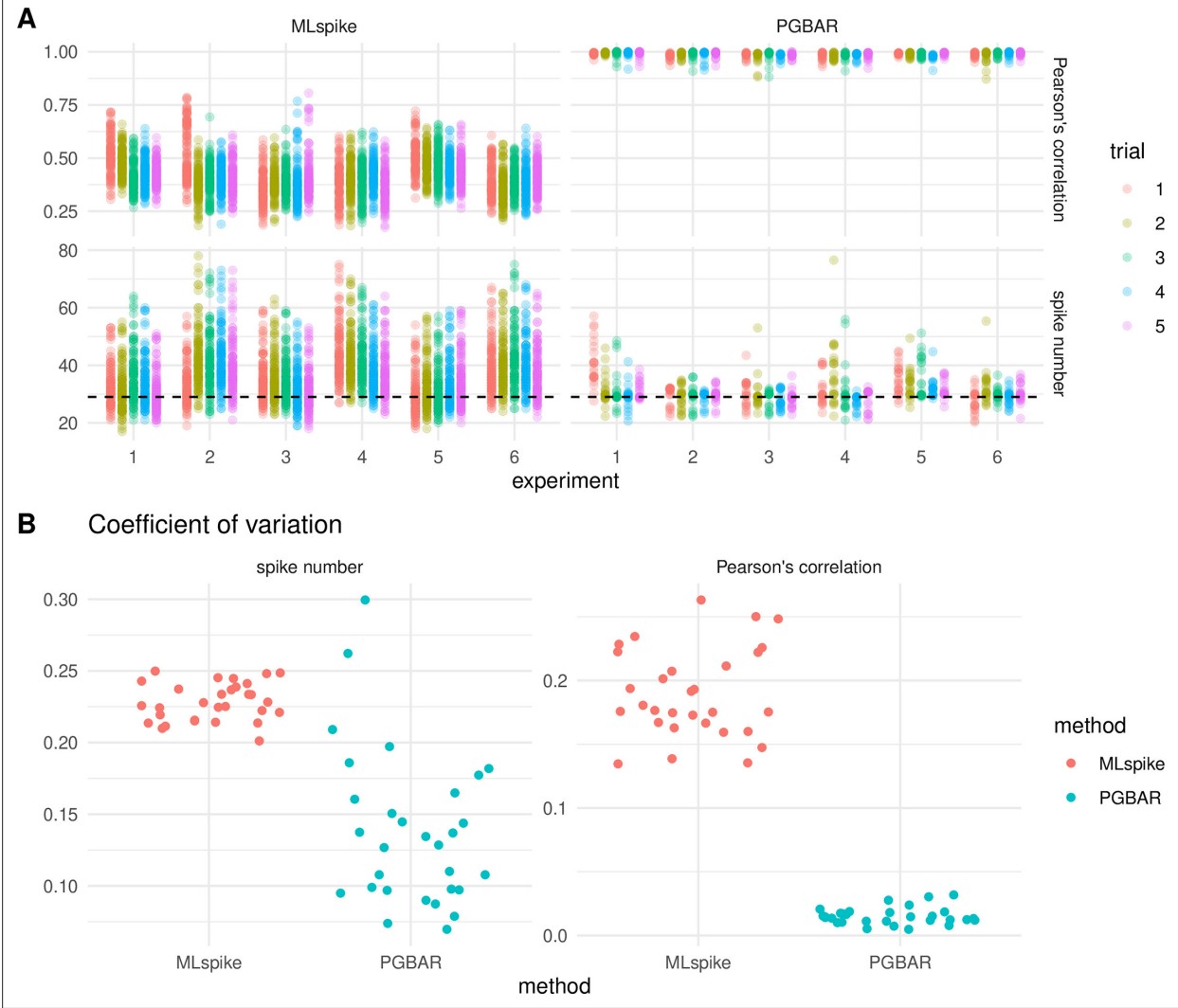

**Figure 10.** Parameter sensitivity in PGBAR and MLspike. (**A**) Pearson's correlation and predicted spike count for PGBAR and MLspike across the same six GC somata (experiment) as in *Figure 9* (five trials, 29 AP per trial) and trials (n=5) with varying PGBAR prior hyperparameters and MLspike model parameters in the range between –50% and +50% from the optimal value. (**B**) Coefficient of variation over parameter sets for both Pearson's correlation and spike number associated to each prediction and comparison between PGBAR and MLspike.

## Discussion

Calcium-sensitive fluorescent indicators are essential tools for monitoring neuronal population activity in model organisms. However, extracting underlying firing patterns from fluorescence time series is challenging due to low signal-to-noise ratio, incomplete knowledge of indicator dynamics, complex firing statistics, and unknown fluorescence modulation. Despite the proliferation of methodologies developed to address this issue, limited attention has been paid to estimating the statistical uncertainties associated with spike inference. The vast majority of spike-detection algorithms rely on optimization techniques and provide only point estimates of detected spikes. Quantifying statistical uncertainties is essential for comparing firing patterns across neurons (*Diana et al., 2019*) and establishing causal relationships. The work of *Pnevmatikakis et al., 2013* and *Vogelstein et al., 2009* addressed this issue by using Monte Carlo methods to approximate the full posterior distributions of spiking patterns. Building upon their work, here we improve the generative model used to infer spiking patterns from fluorescence time series and describe efficient Monte Carlo strategies to estimate spike times and their statistical uncertainties.

## Bursting dynamics and baseline modulation

Neural activity patterns are not always well described by a simple Poisson spiking process. The PGBAR is the first Monte Carlo method (*Pnevmatikakis et al., 2013*; *Vogelstein et al., 2009*; *Greenberg et al., 2018*) to perform statistical inference using a non-homogeneous Poisson firing-and-baseline-modulation model. PGBAR uses a two-state process to enable transitions between low and high firing rates. This feature is used to mimic alternation between periods of low baseline firing and bursting activity transients, during which the firing rate increases significantly. We have shown that neglecting bursting activity in the inference model can lead to biased results, particularly at low SNR and high firing rates. By explicitly modeling bursting dynamics, PGBAR produced unbiased results across all noise and frequency levels when tested on simulated data (*Figure 3*).

Although the model used in *Pnevmatikakis et al., 2013* did not account for fluorescence baseline modulation, the authors acknowledged that baseline fluctuations in in vivo recordings would be essential to consider. PGBAR employs a Gaussian random walk, as in MLSpike (*Deneux et al., 2016*), to model slow changes in baseline fluorescence over the course of the recording. While this is a simple Markov model of fluorescence baseline, it introduces additional noise. To avoid this effect, alternative baseline models, such as the integrated random walk, can be employed.

## Joint estimation of time-independent parameters and dynamic variables

We employed for the first time state-of-the-art particle Gibbs algorithms to infer spikes from noisy fluorescence. This is a key novelty compared with previous SMC-based methods (*Vogelstein et al., 2009*; *Greenberg et al., 2018*), enabling joint estimation of time-independent parameters and dynamic variables. Estimating time-independent model parameters is a well-known challenge in spike detection algorithms, typically requiring ad hoc calibration procedures, grid search, or manual settings. Because spike inference is sensitive to parameters, such as the calcium response amplitude, rise and decay kinetics, and noise level, errors in these parameters can substantially affect the accuracy of spike time estimates. By jointly sampling model parameters and latent variables, PGBAR eliminates the need for separate calibration and ensures that parameter uncertainty is propagated to spike time estimates in a principled manner. As illustrated in *Figure 10*, this yields more robust inference than existing methods, such as MLSpike, which are more sensitive to parameter variation. In addition, PGBAR enables users to calibrate the inference of action potentials by setting prior mean and variance of phenomenological parameters (e.g. rise and decay constants, firing rates, bursting frequencies).

## Comparison with benchmark datasets

The proliferation of spike inference methodologies led to the development of community-based initiatives (*Theis et al., 2016*; *Berens et al., 2018*) to rank the performance of available methods. We applied our approach to the CASCADE dataset (*Rupprecht et al., 2021*), which provides a curated repository of neuronal recordings from mice and zebrafish using various calcium indicators. The performance of PGBAR, measured by the correlation coefficient with ground truth spikes, falls within one quartile of the correlation distributions of existing unsupervised approaches. In addition, it provides information about the statistical uncertainty associated with spike detection that is not currently possible with state-of-the-art techniques. While retraining supervised methods, such as CASCADE on high-frequency or GCaMP8f ground-truth datasets could further improve their performance, limitations in dataset availability and generalization across acquisition regimes motivate complementary, training-free approaches, such as PGBAR.

## PGBAR detection of short high-frequency bursts using an ultrafast calcium indicator

PGBAR employs a second-order autoregressive process to link spiking activity to fluorescence. This simple model captures the basic qualitative aspects of calcium transients and is well suited to linear indicators. For this reason, we have tested the performance of PGBAR on the ultrafast GCaMP8f (*Zhang et al., 2023*), which exhibits improved linearity compared with previous calcium probes. We showed that combining PGBAR with GCaMP8f enables detection of 5 ms inter-spike intervals with an accuracy of 2.5 ms in single trials, thereby providing a statistical tool for estimating high-frequency neural activity patterns.

## PGBAR limitations and future perspectives

Although full Bayesian inference is computationally expensive, SMC algorithms are highly parallelizable. Posterior distributions are represented by particles that are simultaneously propagated through time. In particular, GPU parallelization of SMC methods is an active field of research in computational statistics. Future advances in this direction could significantly enhance these methods and provide tools for online processing of fluorescence time series.

Many commonly used indicators exhibit a nonlinear fluorescence response to increases in intracellular calcium (*Chen et al., 2013*). The autoregressive model used in this study does not account for that nonlinearity. This work provides a statistical framework that can be generalized to more specific biophysical models of calcium indicators to account for non-linear effects. In addition, these models depend on rate constants that are difficult to measure directly. The inability to constrain model parameters is a challenge for spike inference, as different parameter combinations can equally well account for the observed fluorescence by adjusting the inferred spike pattern. For this reason, when model parameters are not identifiable, the spike inference problem from fluorescence time series is ill-defined. Our Bayesian framework offers a systematic approach to address this issue by exploiting current and future data on the kinetics of calcium indicators through informative priors. Thanks to our joint sampling of model parameters and spike patterns, our Monte Carlo method can constrain parameter inference to biophysically relevant ranges.

## Materials and methods

### Particle Gibbs with ancestor sampling

The PGAS step used in Algorithm 1 to sample latent state trajectories was introduced in *Lindsten et al., 2014* to improve the performance of the original particle Gibbs sampler (*Andrieu et al., 2010*). We refer to their original works for details on the method's convergence and mixing properties. The PGAS step is an SMC algorithm that generates a new latent-state trajectory from a reference trajectory and the model parameters. We initialize the algorithm with a set of $N$ latent states (particles) at time $t = 1$,

$$X_1^{(i)} = \{q_1^{(i)}, s_1^{(i)}, C_1^{(i)}, b_1^{(i)}\} \quad i = 1, \cdots, N \tag{14}$$

where the first $N - 1$ are sampled from a proposal distribution equal to the probability of the initial state $\mu^\theta(X_1)$

$$\mu^\theta(X_1) = \rho_1(X_1) = \frac{(r_{q_1}\Delta)^{s_1}}{s_1!} e^{-r_{q_1}\Delta} \cdot \frac{e^{-b_1^2/2}}{\sqrt{2\pi}} \delta^2(C_1 - \hat{C}_1) \tag{15}$$

where bursting and baseline firing states $q_1 = 0, 1$ have equal probability and the calcium vector is constrained by the initial condition of *Equation 5*, $\hat{C}_1 \equiv (c_0 + AS_1, 0)$ and the model parameter $c_0$. The last particle is constrained by the reference trajectory $X'$. To conclude the initialization stage, we assign importance weights $w_1^{(i)}$ to all particles

$$w_1^{(i)} = \mu^\theta(X_1^{(i)}) g^\theta(F_1|X_1^{(i)})/\rho_1(X^{(i)}), \quad i = 1, \cdots, N. \tag{16}$$

Next, for $t > 1$, we evolve the particle system through time by assigning ancestor particles $\{\tilde{X}_{t-1}^{(i)}\}_{i=1}^N$, propagate these to time $t$ to get a new set of particles $\{X_{1:t}^{(i)}\}_{i=1}^N$ and assigning weights $w_t^{(i)}$. For the first $N - 1$ particles, the ancestors are obtained by multinomial resampling from the particle system at time $t - 1$ with probability proportional to the importance weights $w_{t-1}^{(i)}$. The ancestor $J$ of the last particle is drawn from the distribution.

$$\mathbb{P}(J = i) = \frac{w_{t-1}^{(i)} f_t^\theta(X_t'|X_{t-1}^{(i)})}{\sum_{k=1}^N w_{t-1}^{(k)} f_t^\theta(X_t'|X_{t-1}^{(k)})} \tag{17}$$

exploiting the fact that we know its state at time $t$.

In order to evolve the particle system through time, we need to set a proposal distribution $\rho_t(X_t|X_{t-1})$ to sample new latent states. This proposal is then incorporated into the reweighting stage. Although

the choice of the proposal distribution is arbitrary, it can be shown that the conditional distribution $P(X_t|X_{t-1}, F_t)$ reduces the variance of the importance weights. We can express this optimal proposal as

$$P(X_t|F_t, X_{t-1}) = \frac{P(X_t, F_t|X_{t-1})}{\int_{X_t} P(X_t, F_t|X_{t-1})} = \frac{f_t^\theta(X_t|X_{t-1})g_t^\theta(F_t|X_t)}{Z^\theta(X_{t-1}, F_t)} \tag{18}$$

where $Z^\theta(X_{t-1}, F_t)$ is the normalization factor as a function of the latent state at time $t-1$ and current fluorescence $F_t$. We can now use the expressions of $f_t^\theta$ and $g_t^\theta$ in **Equations 11; 12** for our model to compute the optimal proposal distribution. To do so, we decompose $P(X_t|X_{t-1}, F_t)$ as the product

$$P(X_t = \{q_t, s_t, C_t, b_t\}|X_{t-1}, F_t) = P(q_t, s_t|X_{t-1}, F_t) \cdot P(C_t|C_{t-1}, s_t) \cdot P(b_t|q_t, s_t, C_t, X_{t-1}, F_t) \tag{19}$$

where we used the fact that $C_t$ is deterministic and only depends on $C_{t-1}$ and the spike count at time $t$. The idea is to use this chain decomposition to sample first the firing state $q_t$ and the spike count $s_t$, then calculate $C_t$ from its deterministic evolution, and finally sample the baseline $b_t$ from its distribution conditional on the other variables. The first term $P(q_t, s_t|X_{t-1})$ can be obtained by integrating the product $f_t^\theta(X_t|X_{t-1})g_t^\theta(F_t|X_t)$ over $b_t$ and $C_t$ and then normalizing the result. The integration over $b_t$ and $C_t$ leads to

$$
\begin{aligned}
\int db_t\, dC_t f_t^\theta(X_t \mid X_{t-1})\, g_t^\theta(F_t \mid X_t) &= \int db_t\, dC_t\, \delta^{(2)}(C_t - M \cdot C_{t-1} - AS_t) \cdot W_{q_{t-1}q_t} \frac{(r_{q_t}\Delta)^{s_t}}{s_t!} e^{-r_{q_t}\Delta}.\\
&\quad \cdot (2\pi\Delta\sigma_b^2)^{-1/2} \exp\left(-\frac{1}{2\Delta\sigma_b^2}(b_t - b_{t-1})^2\right) \cdot\\
&\quad (2\pi\sigma^2)^{-1/2} \cdot \exp\left[-\frac{1}{2\sigma^2}(F_t - c_t - b_t)^2\right]\\
&= W_{q_{t-1}q_t} \frac{(r_{q_t}\Delta)^{s_t}}{s_t!} e^{-r_{q_t}\Delta} \cdot I(b_{t-1}, F_t - c_t, \Delta\sigma_b^2, \sigma^2)
\end{aligned}
\tag{20}
$$

where we introduced the function $I(y_1, y_2, \sigma_1^2, \sigma_2^2)$ as the integral

$$I(y_1, y_2, \sigma_1^2, \sigma_2^2) = (2\pi\sigma_1^2)^{-1/2}(2\pi\sigma_2^2)^{-1/2} \int dx\, e^{-\frac{(x-y_1)^2}{2\sigma_1^2} - \frac{(x-y_2)^2}{2\sigma_2^2}} = \frac{\exp\left[-\frac{1}{2}\frac{(y_1 - y_2)^2}{\sigma_1^2 + \sigma_2^2}\right]}{\sqrt{2\pi(\sigma_1^2 + \sigma_2^2)}} \tag{21}$$

The normalization factor $Z^\theta(X_{t-1}, F_t)$ is obtained by taking the sum of **Equation 20** over firing state and spike count:

$$Z^\theta(X_{t-1}, F_t) = \sum_{q' \in \{0,1\}} \sum_{s'=0}^{\infty} W_{q_{t-1}q'} \frac{(r_{q'}\Delta)^{s'}}{s'!} e^{-r_{q'}\Delta} \cdot I(b_{t-1}, F_t - c_t, \Delta\sigma_b^2, \sigma^2). \tag{22}$$

To draw a combination of $q_t$ and $s_t$ from this distribution, we applied a cutoff to the number of spikes per time step $S^{(max)} = 20$. We constructed a probability matrix of size $2 \times S^{(max)}$ for all combinations of firing state and spike count.

To obtain the full conditional distribution of $b_t$ we consider again the product $f_t^\theta(X_t|X_{t-1})g_t^\theta(F_t|X_t)$ and by keeping only terms in $b_t$ and normalizing we obtained a Gaussian distribution with mean $\mu_{prop}$ and variance $\sigma_{prop}$ given by

$$\mu_{prop} = \frac{b_{t-1}\sigma^2 + (F_t - c_t)\sigma_b^2\Delta}{\sigma^2 + \sigma_b^2\Delta} \tag{23}$$

$$\sigma_{prop}^2 = \left(\frac{1}{\sigma^2} + \frac{1}{\sigma_b^2\Delta}\right)^{-1}. \tag{24}$$

The final step is to reweight all particles using the importance weight

$$w_t^{(i)} = f_t^\theta(X_t^{(i)}|\tilde{X}_{t-1}^{(i)})g_t^\theta(F_t|X_t^{(i)})/\rho_t(X_t^{(i)}|\tilde{X}_{t-1}^{(i)}) \tag{25}$$

However, due to the form of the optimal proposal in *Equation 18*, the importance weights reduce to the normalization factor calculated in *Equation 22*.

$$w^{(i)} = Z^{\theta}(X_{t-1}^{(i)}, F_t) \tag{26}$$

---

Algorithm 2. **PGAS kernel.**

---

**Input:** Reference trajectory $X'_{1:T}$, and model parameters $\theta$
1: Draw $X_1^{(i)}$ from the poposal distribution $\rho_1$ for $i = 1, \cdots, N-1$
2: Set $X_1^{(N)} = X'_1$
3: Set importance weights $w_1^{(n)} = \mu^{\theta}(X_1^{(n)})g^{\theta}(F_1|X_1^{(n)})/\rho_1(X^{(n)})$ for $n = 1, \cdots, N$
4: **for** $t$ in 2:$T$ **do**
   *// Resampling and ancestor sampling*
5:   Resample $N-1$ particles $\{\tilde{X}_{1:t-1}^{(i)}\}_{i=1}^{N-1}$ with probabilities proportional to the importance weights $\{w_{t-1}^{(i)}\}_{i=1}^{N}$
6:   Draw $J$ with probability $\mathbb{P}(J=i) \propto w_{t-1}^{(i)}f_t^{\theta}(X'_t|X_{t-1}^{(i)})$ and set $\tilde{X}_{1:t-1}^{(N)} = X_{1:t-1}^{(J)}$
   *// Particle propagation*
7:   Draw $X_t^{(i)}$ from the proposal distribution $\rho_t(X_t|\tilde{X}_{t-1}^{(i)})$ for $i = 1, \cdots, N-1$
8:   Set $X_t^{(N)} = X'_t$
9:   Set $X_{1:t}^{(i)} = (\tilde{X}_{1:t-1}^{(i)}, X_t^{(i)})$ for $i = 1, \cdots, N$
   *// Weighting*
10:  Set $w_t^{(i)} = f_t^{\theta}(X_t^{(i)}|\tilde{X}_{t-1}^{(i)})g_t^{\theta}(F_t|X_t^{(i)})/\rho_t(X_t^{(i)}|\tilde{X}_{t-1}^{(i)})$ for $i = 1, \cdots, N$
11: Draw $k$ with $\mathbb{P}(k=i) \propto w_T^{(i)}$
**Output:** $X_{1:T}^{(k)}$

---

## Prior distributions

As discussed in the text, we use a reparameterization of the autoregressive model in terms of the maximal amplitude $A^{(max)}$, rise and decay times $\tau_r$ and $\tau_d$, for which it is easier to design realistic prior distributions based on previous empirical estimates of the kinetics of calcium indicators. We have used truncated normal priors for the maximal amplitude, the initial condition of the autoregressive model $c_0$, the rise and decay time. To calculate the full conditional distributions on bursting/baseline firing rates $r_{0,1}$ and the transition matrix parameters $w_{q \to q'}$ we have used a gamma distribution, whereas for the noise level $\sigma^2$ an inverse gamma distribution.

## Sampling rules for time-independent parameters

In Algorithm 1, after a new latent state trajectory is sampled from the PGAS kernel, we draw time-independent model parameters from the conditional distribution $P(\theta_i|X_{1:T}, F_{1:T})$. We use a mixed approach where the parameters $r_{0,1}$, $w_{q \to q'}$ and $\sigma^2$ are sampled from their full conditional distribution, which can be obtained analytically by using gamma priors. In contrast, kernel parameters, $A^{(max)}$ and $\tau_{r,d}$, are sampled using the Metropolis-Hastings acceptance rule.

The full conditional distribution of a given parameter can be obtained from the joint probability of model parameters, latent state, and fluorescence trajectories

$$P(\theta) \cdot P_{\theta}(X_{1:T}, F_{1:T}|\theta) \tag{27}$$

where $P(\theta)$ is the prior distribution. We will now calculate the full conditionals for firing rates $r_q$, transition parameters $w_{q \to q'}$, and noise variance $\sigma^2$. For simplicity, we will use the same symbols for shape, $\alpha$, and rate, $\beta$, for all prior distributions, although they differ numerically from each other. By combining the expressions in *Equations 10–12* with the gamma prior $\mathrm{gamma}(\alpha, \beta)$ and by keeping only terms proportional to $r_{0,1}$ we obtain

$$P(r_q|\cdots) \propto r_q^{\alpha-1}e^{-\beta r_q - r_q \Delta T} r_q^{\sum_{t:q_t=q} s_t} \tag{28}$$

therefore, the full conditional is a gamma distribution with updated parameters

$$\alpha' = \alpha + \sum_{t:q_t=q} s_t \tag{29}$$

$$\beta' = \beta + \Delta T \tag{30}$$

By applying the same method to the transition rates $w_{q \to q'}$, we have

$$P(w_{q \to q'} | \cdots) \propto w_{q \to q'}^{\alpha - 1} e^{-\beta w_{q \to q'}} w_{q \to q'}^{N_{qq'}} (1 - \Delta w_{q \to q'})^{N_{qq}} \approx w_{q \to q'}^{\alpha + N_{qq'} - 1} e^{-(\beta + \Delta N_{qq})w_{q \to q'}} \tag{31}$$

where the approximation holds when the transition rate between firing states is much slower than the sampling frequency ($w_{q \to q'} \ll \Delta^{-1}$). Therefore, the full conditional is again a gamma distribution with parameters

$$\alpha' = \alpha + N_{qq'} \tag{32}$$

$$\beta' = \beta + \Delta N_{qq} \tag{33}$$

For the noise variance parameter, we used an inverse gamma prior, and by applying the same method, we can compute the full conditional as

$$P(\sigma^2 | \cdots) \propto (\sigma^2)^{-\alpha - 1 - T/2} \exp \left( -\frac{\beta}{\sigma^2} - \frac{1}{2\sigma^2} \sum_t (F_t - c_t - b_t)^2 \right) \tag{34}$$

therefore, the updated shape and rate of the inverse gamma are

$$\alpha' = \alpha + T/2 \tag{35}$$

$$\beta' = \beta + \frac{1}{2} \sum_t (F_t - c_t - b_t)^2 \tag{36}$$

## Response kernel

The response to a single spike can be obtained by writing **Equation 3** in the form of a first-order Markov process in terms of the new variables $C_t \equiv [c_t, c_{t-1}]$ and $S_t \equiv [s_t, 0]$ so that

$$C_t = M \cdot C_{t-1} + A S_t, \quad M = \begin{bmatrix} \gamma_1 & \gamma_2 \\ 1 & 0 \end{bmatrix} \tag{37}$$

with the initial condition $C_1 = [c_0 + A S_1, 0]$. We can now write the solution at time $t$ as

$$C_t = M^{t-1} C_1 + A \sum_{k=2}^{t} M^{t-k} S_k. \tag{38}$$

If $s_t = \delta_{t,1}$ and $c_0 = 0$ then **Equation 38** simplifies to

$$C_t = A M^{t-1} \begin{bmatrix} 1 \\ 0 \end{bmatrix}. \tag{39}$$

By introducing eigenvectors and eigenvalues of $M$

$$\gamma_\pm \equiv \frac{\gamma_1 \pm \sqrt{\gamma_1^2 + 4\gamma_2}}{2}, \quad e_\pm \equiv \begin{bmatrix} \gamma_\pm \\ 1 \end{bmatrix} \tag{40}$$

we can express **Equation 39** as

$$C_t = A M^{t-1} \left( \frac{e_+ - e_-}{\gamma_+ - \gamma_-} \right) = A \left( \frac{\gamma_+^{t-1} e_+ - \gamma_-^{t-1} e_-}{\gamma_+ - \gamma_-} \right) \tag{41}$$

therefore, we have

$$c_t = A \left( \frac{\gamma_+^t - \gamma_-^t}{\gamma_+ - \gamma_-} \right). \tag{42}$$

By setting the time derivative of $c_t$ to zero, we obtain the time to reach the maximal response $\tau_r$ as

$$\tau_r = \frac{\log\left(\frac{g_+}{g_-}\right)}{g_- - g_+}, \quad g_\pm = \log\gamma_\pm \tag{43}$$

whereas if we take the long time limit of *Equation 39* we obtain

$$c_t = A\gamma_+^t[1 - (\gamma_-/\gamma_+)^t] \approx Ae^{t/\tau_d}, \quad \tau_d = -\frac{1}{g_+} \tag{44}$$

### Reparameterization

To parameterize the autoregressive model in terms of kinetic parameters, we need to find the inverse map $\tau_{r,d} \to \gamma_{1,2}$ to obtain the original autoregressive parameters given rise and decay times. By combining the expressions of $\tau_r$ and $\tau_d$ in terms of $g_\pm$, we have

$$\frac{\tau_r}{\tau_d} = \frac{\log\frac{g_-}{g_+}}{\frac{g_-}{g_+} - 1} \tag{45}$$

which shows that the ratio $g_-/g_+$ can be expressed as

$$\frac{g_-}{g_+} = f^{-1}\left(\frac{\tau_r}{\tau_d}\right), \quad f(x) = \frac{\log(x)}{x-1} \tag{46}$$

where the inverse function $f^{-1}(x)$ can be determined by numerical interpolation in the range $[0,1]$. To obtain the original autoregressive parameteres $\gamma_{1,2}$, we first obtain $g_\pm$ as

$$g_+ = -\frac{1}{\tau_d} \tag{47}$$

$$g_- = g_+ \cdot f^{-1}\left(\frac{\tau_r}{\tau_d}\right) \tag{48}$$

and then

$$\gamma_1 = e^{g_+} + e^{g_-} \tag{49}$$

$$\gamma_2 = -e^{g_+ + g_-} \tag{50}$$

### Runtime benchmark

We conducted a thorough runtime benchmarking analysis to evaluate the performance of PGBAR across varying numbers of particles in the PGAS kernel and time-series lengths. Specifically, we measured the execution time of a single iteration of Algorithm 1 to draw a sample of time-dependent variables and the model parameters. The runtime was averaged over 100 posterior samples. We performed this test by varying the number of particles from 50 to 1000, using time series lengths ranging from 200 to 1000 time steps (*Figure 11*). The time to draw a single iteration of PGBAR ranges from a minimum of 40 ms, to process a time series of 200 time steps with 50 PGAS particles, up to 2.3 s to process 1000 time steps with 1000 particles. Our analysis demonstrates that the runtime scales linearly both in the number of PGAS particles and the time series length. All tests were executed on a Dell XPS 15 laptop with CPU 11th Gen Intel(R) Core(TM) i7-11800H @ 2.30 GHz and 32 Gb of RAM.

### Summary statistics

All boxplots follow the standard convention: the box represents the interquartile range, the central line indicates the median, and the whiskers extend to the most extreme values within 1.5 times the interquartile range.

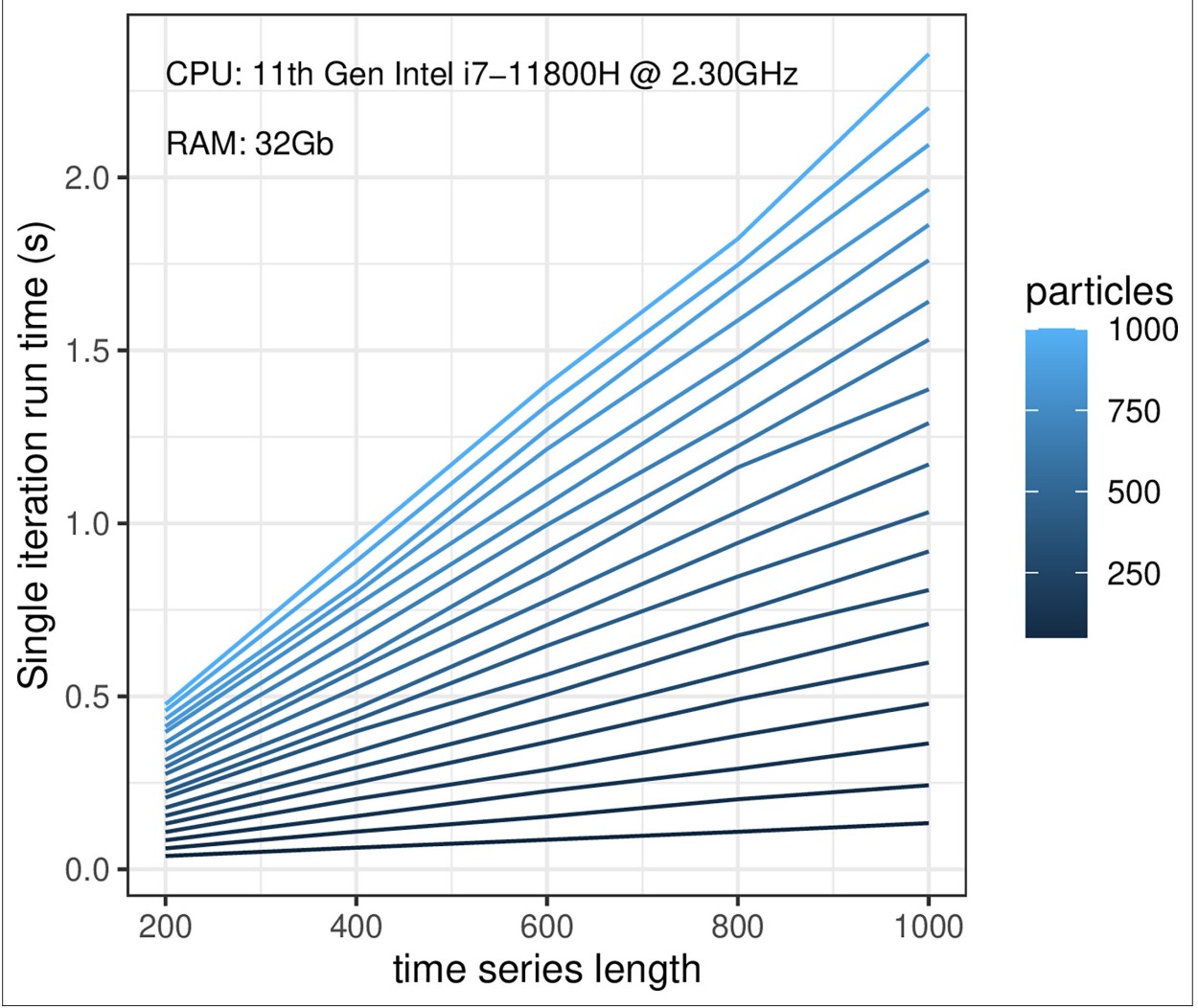

**Figure 11.** Runtime benchmarking. Execution time of a single iteration of Algorithm 1 sampling both model parameters and time-dependent variables. The performance was estimated by averaging over 100 iterations. All tests were conducted on a Dell XPS 15 laptop with a 11[th] Gen Intel(R) Core(TM) i7-11800H @ 2.30 GHz and 32 Gb of RAM.

## Experimental methods

### GCaMP8f virus injection in the Crus I lobule of the cerebellum

Virus injection was targeted to Crus I (6.00 mm posterior to the Bregma, 3.00 mm lateral to the midline; vertical depth of 0.50 mm from pial surface) under deep isofluorane anesthesia. 100 nl of adeno-associated virus encoding GCaMP8f (AAV-DJ-CAG-FLEx-jGCaMP8f, Janelia Research Campus) was injected with Nanoject III (Drummond Scientific) using thin glass pipette (diameter 30 $\mu$m). Injection was made when the C57BL/6J-Gabra6tm2(cre)Wwis mice were 7 weeks old. After injections, the mice were returned to their home cage for 6 weeks to allow time for expression. All experiments were approved by the Ethics Committee 89 of Institut Pasteur (CETEA; protocol approval DHA180006).

### Slice preparation

Acute coronal slices (200 $\mu$m) of the Crus I lobule of the cerebellum was prepared from adult C57BL/6J-Gabra6tm2(cre)Wwis mice, aged 97 and 102 days. Following transcardial perfusion with an ice-cold solution containing (in mM): 2.5 KCl, 0.5 CaCl2, 4 MgCl2, 1.25 NaH2PO4, 24 NaHCO3, 25 glucose, 230 sucrose, and 0.5 ascorbic acid, the brains were removed and placed in the same solution. The solution was bubbled with 95% O2 and 5% CO2. Slices were cut from the dissected lateral cerebellum using a vibratome (Leica VT1200S), and incubated at room temperature for 30 min in a solution containing (in

mM): 85 NaCl, 2.5 KCl, 0.5 CaCl2, 4 MgCl2, 1.25 NaH2PO4, 24 NaHCO3, 25 glucose, 75 sucrose, and 0.5 ascorbic acid. Slices were then transferred to an external recording solution containing (in mM): 125 NaCl, 2.5 KCl, 1.5 CaCl2, 1.5 MgCl2, 1.25 NaH2PO4, 24 NaHCO3, 25 glucose and 0.5 ascorbic acid, and maintained at room temperature for up to 6 hr.

## Cellular imaging

The brain slices containing GCaMP8f-expressing cells were identified with a 4 x objective lens (Olympus UplanFI 4 x, 0.13 NA) using brief illumination with 470 nm light to excite GCaMP8f fluorescence. GCs were identified using infrared Dot-Gradient contrast and a QlClick digital CCD camera (QImaging, Surrey, BC, Canada) mounted on an Ultima multiphoton microscopy system (Bruker Nano Surfaces Division, Middleton, WI, USA) that was mounted on an Olympus BX61W1 microscope, equipped with a water-immersion objective (Olympus 60 x, 1.1 NA). Two-photon excitation was performed with a Ti-sapphire laser (Spectraphysics). To visualize GCs expressing GcaMP8f, two-photon excitation was performed at 920 nm. Infrared Dodt-gradient contrast was used to position the stimulation pipette in the molecular layer, targeting PFs to activate GC bodies directly. Linescan imaging of GC bodies was performed by scanning the entire cell membrane in freehand linescan mode (Prairie View). Total laser illumination per sweep lasted 2000 ms. Fluorescence was detected using both proximal epifluorescence and substage photomultiplier tube gallium arsenide phosphide (H7422PA-40 SEL, Hamamatsu).

## Code and data

In this work, we employed the publicly available CASCADE dataset in *Rupprecht et al., 2021*. Linescan imaging data and the source code of PGBAR are available through the GitHub repository https://github.com/giovannidiana/pgbar, copy archived at *Diana, 2023*.

## Acknowledgements

This project has received funding from the European Union's Horizon 2020 research and innovation programme under the Marie Skłodowska-Curie grant agreement No 896051. This work was also supported by a Pasteur-Roux-Cantarini fellowship of the Institut Pasteur. and ANR 19 CE16 0019, ANR 21 CE16 0036, ANR 23 CE23 0034, and FRM EQU202003010555. We would like to thank Peter Rupprecht for his help with the analysis of the CASCADE dataset and for thorough discussions. We also thank Nicolas Chopin for suggesting the use of backward steps methods in particle Gibbs, Andrea Giovannucci, Marco Banterle, Diana Passaro, and Fredrik Lindsten for helpful discussions.

## Additional information

### Funding

| Funder | Grant reference number | Author |
|---|---|---|
| Horizon 2020 Framework Programme | 896051 | Giovanni Diana |
| Institut Pasteur | | B Semihcan Sermet |
| Agence Nationale de la Recherche | ANR 19 CE16 0019 | David A DiGregorio |
| Agence Nationale de la Recherche | ANR 21 CE16 0036 | David A DiGregorio |
| Agence Nationale de la Recherche | ANR 23 CE23 0034 | David A DiGregorio |
| Fondation pour la Recherche Médicale | EQU202003010555 | David A DiGregorio |

The funders had no role in study design, data collection and interpretation, or the decision to submit the work for publication.

## Author contributions

Giovanni Diana, Conceptualization, Data curation, Software, Formal analysis, Funding acquisition, Validation, Investigation, Visualization, Methodology, Writing – original draft, Project administration, Writing – review and editing; B Semihcan Sermet, Data curation, Validation, Writing – original draft, Writing – review and editing; Gerard J Broussard, Validation; Samuel S-H Wang, Supervision; David A DiGregorio, Resources, Supervision, Funding acquisition, Writing – original draft, Project administration, Writing – review and editing

## Author ORCIDs

Giovanni Diana ⓘ https://orcid.org/0000-0001-7497-5271
B Semihcan Sermet ⓘ https://orcid.org/0000-0002-9983-3605
Samuel S-H Wang ⓘ https://orcid.org/0000-0002-0490-9786
David A DiGregorio ⓘ https://orcid.org/0000-0002-6417-4566

## Ethics

All experiments were approved by the Ethics Committee 89 of Institut Pasteur (CETEA; protocol approval DHA180006).

Reviewer #1 (Public review): https://doi.org/10.7554/eLife.94723.4.sa1
Reviewer #2 (Public review): https://doi.org/10.7554/eLife.94723.4.sa2
Author response https://doi.org/10.7554/eLife.94723.4.sa3

---

# Additional files

## Supplementary files

MDAR checklist

## Data availability

Linescan imaging data and the source code of PGBAR are available through the GitHub repository https://github.com/giovannidiana/pgbar (copy archived at *Diana, 2023*). High-frequency linescan recordings of cerebellar granule cells have been deposited at https://osf.io/t3wm6.

The following dataset was generated:

| Author(s) | Year | Dataset title | Dataset URL | Database and Identifier |
|---|---|---|---|---|
| Diana G, Sermet BS, Broussard GJ, Wang SSH, DiGregorio DA | 2026 | High frequency recordings from cerebellar granule cells | https://osf.io/t3wm6 | Open Science Framework, t3wm6 |

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
