## [Editor Report · eLife Assessment]

This study presents a **valuable** contribution by introducing a model-based, Bayesian method for inferring action potentials from calcium imaging data that directly quantifies uncertainty in spike timing through posterior distributions. Using a Monte Carlo particle Gibbs sampling approach, the method achieves temporal resolution and accuracy comparable to existing techniques while offering the key added benefit of principled uncertainty estimates. The underlying methodology and characterization are **convincing**, and the work will be of particular interest to theoretically oriented neuroscientists seeking rigorous new tools for data-driven parameter inference.

---

## [Referee Report · Reviewer #1 (Public review)]

Summary:

In this study, Diana et al. present a Monte Carlo-based method to perform spike inference from calcium imaging data. A particular strength of their approach is that they can estimate not only averages but also uncertainties of the modeled process. The authors focus on the quantification of spike time uncertainties in simulated data and in data recorded with high sampling rate in cebellar slices with GCaMP8f, and they demonstrate the high temporal precision that can be achieved with their method to estimate spike timing.

Strengths:

- The author provide a solid ground work for sequential Monte Carlo-based spike inference, which extends previous work of Pnevmatikakis et al., Greenberg et al. and others.

- The integration of two states (silence vs. burst firing) seems to improve the performance of the model.

- The acquisition of a GCaMP8f dataset in cerebellum is useful and helps make the point that high spike time inference precision is possible under certain conditions.

Weaknesses:

- Although the algorithm is compared (in the revised manuscript) to other models to infer individual spikes (e.g., MLSpike), these comparisons could be more comprehensive. Future work that benchmarks this and other algorithms under varying conditions (e.g., noise levels, temporal resolution, calcium indicators) would help assess and confirm robustness and useability of this algorithm.

- The mathematical complexity underlying the method may pose challenges for experimentalist who may want to use the methods for their analyses. While this is not a weakness of the approach itself, this highlights the need for further validation and benchmarking in future work, to build user confidence.

Comments on revisions:

Thank you for addressing the final comments, and congrats on this study!

---

## [Referee Report · Reviewer #2 (Public review)]

Summary:

Methods to infer action potentials from fluorescence-based measurements of intracellular calcium dynamics are important for optical measurements of activity across large populations of neurons. The variety of existing methods can be separated into two broad classes: (a) model-independent approaches that are trained on ground truth datasets (e.g., deep networks), and (b) approaches based on a model of the processes that link action potentials to calcium signals. Models usually contains parameters describing biophysical variables, such as rate constants of the calcium dynamics and features of the calcium indicator. The method presented here, PGBAR, is model-based and uses a Bayesian approach. A novelty of PGBAR is that static parameters and state variables are jointly estimated using particle Gibbs sampling, a sequential Monte Carlo technique that can efficiently sample the latent embedding space.

Strengths:

A main strength of PGBAR is that it provides probability distributions rather than point estimates of spike times. This is different from most other methods and may be an important feature in cases when estimates of uncertainty are desired. Another important feature of PGBAR is that it estimates not only the state variable representing spiking activity, but also other variables such as baseline fluctuations and stationary model variables, in a joint process. PGBAR can therefore provide more information than various other methods. The information in the github repository is well-organized. The authors demonstrate convincingly that PGBAR can resolve inter-spike intervals in the range of 5 ms using fluorescence data obtained with a very fast genetically encoded calcium indicator at very high sampling rates (line scans at >= 1 kHz).

Weaknesses:

The accuracy of spike train reconstructions is not higher than that of other model-based approaches, and lower than the accuracy of a model-independent approach based on a deep network in a regime of commonly used acquisition rates.

Comments on revisions:

I have no further comments on the manuscript.

---

## [Author Response]

The following is the authors’ response to the previous reviews

**Public Reviews:**

**Reviewer #1 (Public review):**
Summary:In this study, Diana et al. present a Monte Carlo-based method to perform spike inference from calcium imaging data. A particular strength of their approach is that they can estimate not only averages but also uncertainties of the modeled process. The authors focus on the quantification of spike time uncertainties in simulated data and in data recorded with high sampling rate in cebellar slices with GCaMP8f, and they demonstrate the high temporal precision that can be achieved with their method to estimate spike timing.Strengths:- The author provide a solid ground work for sequential Monte Carlo-based spike inference, which extends previous work of Pnevmatikakis et al., Greenberg et al. and others.- The integration of two states (silence vs. burst firing) seems to improve the performance of the model.- The acquisition of a GCaMP8f dataset in cerebellum is useful and helps make the point that high spike time inference precision is possible under certain conditions.Weaknesses:- Although the algorithm is compared (in the revised manuscript) to other models to infer individual spikes (e.g., MLSpike), these comparisons could be more comprehensive. Future work that benchmarks this and other algorithms under varying conditions (e.g., noise levels, temporal resolution, calcium indicators) would help assess and confirm robustness and useability of this algorithm.

The metrics used for comparison follow the field's benchmarking conventions (see the CASCADE paper, Rupprecht et al. 2021). Indeed, improved standardized methods would be ideal to develop, which is beyond the scope of this manuscript.

- The mathematical complexity underlying the method may pose challenges for experimentalist who may want to use the methods for their analyses. While this is not a weakness of the approach itself, this highlights the need for further validation and benchmarking in future work, to build user confidence.

We acknowledge the challenges of understanding the mathematics underlying our method, but such a study is necessary to ensure its accuracy and reliability. Indeed, we will strive to improve the technique's user-friendliness in future instantiations.

**Reviewer #2 (Public review):**
Summary:Methods to infer action potentials from fluorescence-based measurements of intracellular calcium dynamics are important for optical measurements of activity across large populations of neurons. The variety of existing methods can be separated into two broad classes: (a) model-independent approaches that are trained on ground truth datasets (e.g., deep networks), and (b) approaches based on a model of the processes that link action potentials to calcium signals. Models usually contains parameters describing biophysical variables, such as rate constants of the calcium dynamics and features of the calcium indicator. The method presented here, PGBAR, is model-based and uses a Bayesian approach. A novelty of PGBAR is that static parameters and state variables are jointly estimated using particle Gibbs sampling, a sequential Monte Carlo technique that can efficiently sample the latent embedding space.Strengths:A main strength of PGBAR is that it provides probability distributions rather than point estimates of spike times. This is different from most other methods and may be an important feature in cases when estimates of uncertainty are desired. Another important feature of PGBAR is that it estimates not only the state variable representing spiking activity, but also other variables such as baseline fluctuations and stationary model variables, in a joint process. PGBAR can therefore provide more information than various other methods. The information in the github repository is well-organized.Weaknesses:On the other hand, the accuracy of spike train reconstructions is not higher than that of other model-based approaches, and clearly lower than the accuracy of a model-independent approach based on a deep network. The authors demonstrate convincingly that PGBAR can resolve inter-spike intervals in the range of 5 ms using fluorescence data obtained with a very fast genetically encoded calcium indicator at very high sampling rates (line scans at >= 1 kHz).

In the revision, Figure 9 shows that temporal accuracy is very similar between PGBAR and the supervised method, CASCADE, and that PGBAR has a lower false positive rate. These results support the effectiveness of unsupervised Monte Carlo sampling, even with a simple autoregressive model.

**Recommendations for the authors:**

**Reviewer #1 (Recommendations for the authors):**
I'd like to thank the authors for their revisions. Their comments have addressed all my concerns, and I thank them for the clarifications. I have no further comments, except a few minor notes that the authors may consider or not:- The paragraph starting in line 367 is newly written and not yet as clear and mature as other parts of the manuscript. It is at several sentences roughly clear what it is about, but the precision of the wording is lacking. For example "distributions of the average time from ground-truth" seems a bit unclear, maybe "distributions of the average time of estimate spikes from ground-truth spikes" instead. Similarly, "the false detection rate, defined as the difference between detected and ground-truth spikes ..." could be rephrased using the difference between "numbers of spikes" instead of the difference between "spikes". But all of this is minor.- In the new Figure 9A, the error bars for the MLSpike method seem to be absent. In the same figure legend, it should be "excess" instead of "excess".

We thank the reviewer for the feedback. We revised the wording of the new paragraph in response to the reviewer’s suggestions, restored the missing error bar in Figure 9, and corrected the figure legend.

**Reviewer #2 (Recommendations for the authors):**
Comparison to CASCADE: as far as I know there are no CASCADE models that have been trained on ground truth data in the regime of very fast (line scan) sampling, which is rarely used. A fair comparison of spike time estimates between PGBAR and CASCADE should take this into account. This can be done by training a new CASCADE model using the dataset of this paper. Given that performance of PGBAR and CASCADE is very similar already now (except for the false positive rate), a CASCADE model optimized for high sampling rate may be expected to catch up with (or even exceed) the performance of PGBAR. At a minimum, this possibility should be discussed.

While this may be true, retraining a CASCADE model on high-frequency ground-truth data is beyond the scope of this manuscript. Indeed, a retrained CASCADE model optimized for line-scan or GCaMP8f data could improve performance and potentially match or exceed PGBAR, particularly in reducing false positives.

Our aim, however, is not to benchmark supervised methods under their optimal retraining conditions, but to provide an unsupervised alternative that does not rely on labeled training data. In practice, retraining supervised models is constrained by the availability of suitable ground-truth datasets and by the uncertainty in how the method generalizes to acquisition regimes that differ substantially from the training set.

We have therefore added a sentence in the Discussion (at the end of the subsection Comparison with benchmark datasets):

[...] “While retraining supervised methods such as CASCADE on high-frequency or GCaMP8f ground-truth datasets could further improve its performance, limitations in dataset availability and generalization across acquisition regimes motivate complementary, training-free approaches such as PGBAR.”

As stated in the manuscript, future extensions, such as using nonlinear biophysical models as the generative model for Monte Carlo–based inference, may further improve spike estimation accuracy.